# FEATURE MODULATING FOR DIFFUSION MODELS

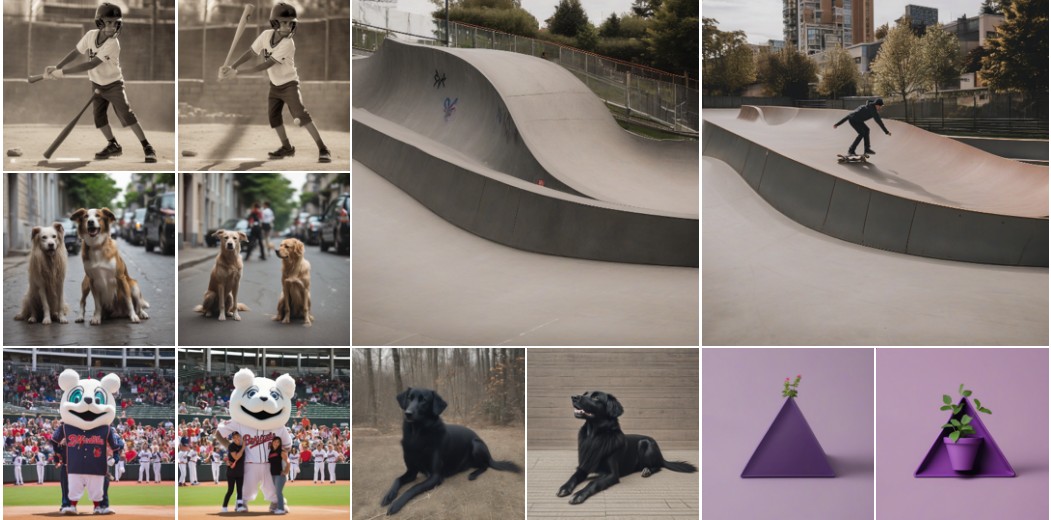

Figure 1: Comparison of generated images before and after applying our ***Feature Modulating (FM)*** method. In each image pair, the left image is generated by SDXL (Podell et al., 2023), while the right image shows our enhanced result.

## ABSTRACT

We propose *Feature Modulating (FM)*, a training-free approach that enables image quality improvement and better text-image alignment in text-to-image diffusion models. Rather than relying on additional input information, our heuristic FM module directly modulates latent features during the denoising steps. The modulation alters the feature distribution, leading to differences in the generated images. To explore the impact of feature modulation, we introduce a channel-wise monotonic modulation function that adjusts feature values using a single parameter, facilitating the obtainment of high quality images. The FM module is architecture-agnostic and can be integrated into existing diffusion models. Extensive experiments across multiple benchmarks demonstrate the ability of our feature modulation to enhance image quality, semantic fidelity and realism. The code will be made publicly available upon acceptance.

## 1 INTRODUCTION

Generative models such as Variational Autoencoders (VAEs) (Kingma & Welling, 2013; Wang et al., 2021), Generative Adversarial Networks (GANs) (Goodfellow et al., 2014; Mirza & Osindero, 2014; Brock et al., 2018; Karras et al., 2018; 2019; 2020; 2021), and vector-quantized frameworks (Van Den Oord et al., 2017; Esser et al., 2021b) have received substantial attention for their ability to produce high-quality images. In contrast, diffusion probabilistic models (Ho et al., 2020; Dhariwal & Nichol, 2021; Rombach et al., 2022; Ramesh et al., 2022; Esser et al., 2021a; Gu et al., 2022; Nichol et al., 2021; Saharia et al., 2022; Gal et al., 2023; Kumari et al., 2022; Kawar et al., 2022) represent a more recent class of generative methods that has rapidly risen to prominence. These models leverage a fixed Markov chain to methodically inject and then remove Gaussian noise, allowing them to capture intricate structural details and achieve state-of-the-art results in various tasks

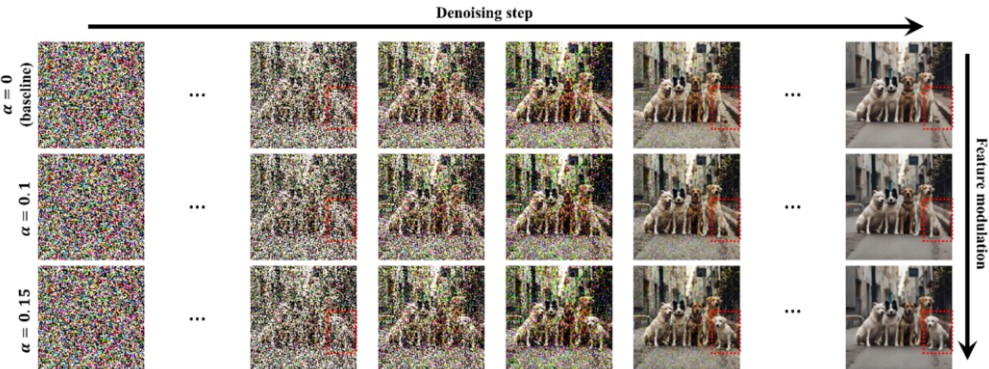

Figure 2: Denoising process under different feature modulation parameters. Given the prompt *"Five dogs on the street."*, the figure illustrates how feature modulation affects the denoising process. As highlighted by the red dashed boxes, feature modulation alters the model's semantic interpretations of foreground and background, resulting in different denoising trajectories.

such as image synthesis, video generation, and molecule design. Recent advancements in diffusion-based approaches have nonetheless brought remarkable improvements on image synthesis (Ho et al., 2020; Rombach et al., 2022; Saharia et al., 2022; Esser et al., 2024), in both the detail and diversity of generated outputs, enabling innovations.

Diffusion models consist of two key procedures: a *diffusion process*, in which Gaussian noise is progressively added to an input until it resembles pure noise, and a *denoising process*, in which the original sample is restored through iterative noise removal. A U-Net architecture (Ho et al., 2020) is typically employed to predict the noise at each denoising step. More recently, latent diffusion methods have replaced the standard U-Net backbone with a transformer operating on latent patches (Chen et al., 2023; Peebles & Xie, 2023; Esser et al., 2024; Chen et al., 2024), extending these principles to a transformer-based denoising framework. Despite the growing success and scale of diffusion models, their outputs still struggle with complex or lengthy prompts, as illustrated in the left image of Fig. 2Fig. 1. While recent works have improved controllability through specialized conditioning mechanisms, such as spatial guidance (Zhang & Agrawala, 2023; Mou et al., 2023), these approaches typically require auxiliary inputs (sketches, depth maps, or image/video prompts). For text-only scenarios without auxiliary inputs, reliably adhering to complex linguistic constraints remains challenging. This gap persists despite architectural innovations, which aim to better ground semantic relationships.

To fill the gap, we aim to provide a modulating module which has the ability to improve the generated image quality for prompts that current text-to-image diffusion models are struggle with. Our approach leverages the relationship between latent features and generated images. We assume that modifying specific channels of the latent feature has meaningful impact on the attributes of generated images. The results in Fig. 2 shows that manipulating the feature distribution during the denoising process could lead to noticeable changes in the generated image.

In this work, we propose a training-free *Feature Modulating* (FM) strategy for diffusion models. Our approach adopts a channel-wise function to modulate features during the denoising process, which consequently alters the generated images. This function is carefully designed to allow flexible control while preserving the feature monotonicity. We introduce a quadratic function with a single parameter to allow customized adjustment. Our FM is promising to enhance image quality by selecting part of the channels to be modulated and tuning the parameter. Our FM does not rely on a specific architecture, making it suitable for existing diffusion models. The effectiveness of our FM is validated by experiments on three diffusion models across several benchmarks. Our main contributions are summarized as follows:

- We investigate the potential of feature modulating at channel level in diffusion models, revealing that manipulating in-channel feature distribution can lead to a significant influence on the output images.

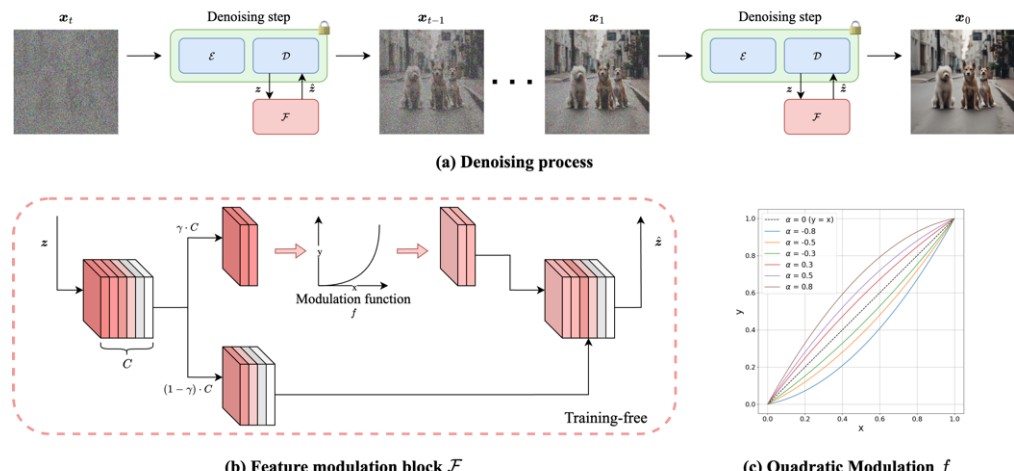

Figure 3: Overview of the proposed method. (a) Denoising process, where the original encoder-decoder of SD is frozen, and our FM is applied in a training-free manner. (b) Feature modulation block, where a proportion $\gamma$ of feature channels are modulated. (c) Modulation functions under different $\alpha$ values.

- We introduce a simple yet effective Feature Modulating (FM) scheme, which boosts denoising performance by adjusting the latent feature maps during denoising steps, requiring no extra training.
- Our FM framework integrates effortlessly into existing Stable Diffusion variants, demonstrating notable improvements in image quality for various diffusion-based methods.

## 2 RELATED WORK

**Text-to-Image Diffusion Models.** Diffusion models have become the leading approach for text-to-image synthesis, typically using a U-Net to iteratively denoise latent representations (Ho et al., 2020; Rombach et al., 2022; Ramesh et al., 2022). Stable Diffusion pioneered operating in a lower-dimensional latent space to cut sampling steps without sacrificing quality (Podell et al., 2023). Transformer-based DiT (Peebles & Xie, 2023) further replaces the U-Net for improved scalability and multimodal alignment, while recent variants such as PixArt-$\alpha$ (Chen et al., 2023) and Stable Diffusion 3 (Esser et al., 2024) focus on enhanced controllability and image fidelity.

**Control for Diffusion Models.** Recent works augment diffusion models with additional conditioning to improve controllability: ControlNet (Zhang & Agrawala, 2023) enables fine-grained guidance without merging, and X-Adapter (Ran et al., 2023) reuses pretrained adapters for larger models. MS-Diffusion (Wang et al., 2025) and Ctrl-Adapter (Lin et al., 2024) integrate multiple conditions for diverse control. In DiT-based architectures, PixArt-$\delta$ (Chen et al., 2024) focuses on practical content creation, while OminiControl (Tan et al., 2024) and RelaCtrl (Cao et al., 2025) incorporate image-conditioned signals. Training-free methods such as FreeU (Si et al., 2024) and RB-Modulation (Rout et al., 2024) enhance output quality without retraining, and specialized techniques like Make-it-Count (Binyamin et al., 2025) and Attend-and-excite (Chefer et al., 2023) offer control over numeric accuracy and object replacement.

## 3 METHODOLOGY

### 3.1 PRELIMINARIES

Diffusion models, such as Denoising Diffusion Probabilistic Models (DDPM) (Ho et al., 2020), generate data by transforming noise into meaningful samples through a two-stage process. In the first stage, known as the forward diffusion process, a clean sample $x_0 \sim q(x_0)$ is progressively corrupted over $T$ timesteps by adding Gaussian noise. At each timestep $t$, the sample $x_{t-1}$ is

perturbed according to a Gaussian distribution with a variance defined by a schedule $\beta_1, \ldots, \beta_T$, such that:

$$q(\boldsymbol{x}_t \mid \boldsymbol{x}_{t-1}) = \mathcal{N}\Big(\boldsymbol{x}_t; \sqrt{1 - \beta_t}\,\boldsymbol{x}_{t-1},\, \beta_t\mathbf{I}\Big). \tag{1}$$

In the second stage, the reverse (denoising) process seeks to recover the original data by iteratively removing the added noise. This is accomplished by approximating the conditional distribution of $\boldsymbol{x}_{t-1}$ given the current noisy observation $\boldsymbol{x}_t$. Neural networks are employed to predict the mean $\boldsymbol{\mu}_\theta(\boldsymbol{x}_t, t)$ and covariance $\boldsymbol{\Sigma}_\theta(\boldsymbol{x}_t, t)$ of the reverse transition:

$$p_\theta(\boldsymbol{x}_{t-1} \mid \boldsymbol{x}_t) = \mathcal{N}\Big(\boldsymbol{x}_{t-1}; \boldsymbol{\mu}_\theta(\boldsymbol{x}_t, t),\, \boldsymbol{\Sigma}_\theta(\boldsymbol{x}_t, t)\Big). \tag{2}$$

By sequentially denoising the sample from $\boldsymbol{x}_T$ down to $\boldsymbol{x}_0$, the model is able to reconstruct high-fidelity images from noise.

### 3.2 FEATURE MODULATION

For clarity and convenience in presentation, we conceptually decompose a single denoising step $p_\theta(\cdot)$ into two distinct blocks based on operations before and after feature modification as shown in Fig. 3: an Encoder Block $\mathcal{E}$, which represents the feature extraction (encoding-like) process, and a Decoder Block $\mathcal{D}$, responsible for noise prediction and refinement (decoding-like). Specifically, during a single denoising step at time $t$, the encoder block $\mathcal{E}$ maps the noisy input $\boldsymbol{x}_t$ into a latent feature space, yielding a feature map $\boldsymbol{z} \in \mathbb{R}^{C \times H \times W}$ by $\boldsymbol{z} = f_{\text{enc}}(x_t, t; \theta_\mathcal{E})$. Subsequently, the decoder block $\mathcal{D}$ takes the modified latent features $\hat{\boldsymbol{z}}$ and maps them back to the pixel space to generate the denoised image by $\hat{\epsilon} = f_{\text{dec}}(\hat{\boldsymbol{z}}, t; \theta_\mathcal{D})$, where $\hat{\epsilon}$ is the predicted noise to be removed from $x_t$. We denote our feature modulation block with $\mathcal{F}$ in Fig. 3.

Our hypothesis is that adjusting the feature distribution at specific levels of the denoising process can alter the generated image. Our desired FM function is expected to meet the following four primary objectives: 1) Each channel is modulated independently to avoid inter-channel correlations; 2) Each channel's modulated values must remain in the original range to avoid introducing artifacts; 3) The function should be monotonous, preserving relative differences among neighboring feature values; 4) The modulation depends on the feature values themselves and a single parameter $\alpha$. We denote FM function as $f_M(\cdot)$, that modulates the feature maps by $\hat{\boldsymbol{z}} = f_M(\boldsymbol{z}, t; \alpha)$. In the following, we explain the application of our FM function in a single denoising step. Unless otherwise stated, we use one global scalar $\alpha$ shared across all denoising timesteps in sampling; more general timestep-dependent schedules are also possible and are visualized in Appendix F (Fig. F11).

#### 3.2.1 CHANNEL-WISE FEATURE MODULATION PROCESS

To satisfy these four requirements, we first normalize each channel to the range of $(0, 1]$, then apply a quadratic function with a coefficient $\alpha$, and finally remap the feature back to the original range.

**Normalization.** Let $\boldsymbol{z}_c$ denote the feature of a particular channel $c$, where $c \in \{1, 2, \ldots, C\}$ indexes the channels. The normalization for feature $\boldsymbol{z}_c$ is

$$\widetilde{\boldsymbol{z}}_c = \frac{\boldsymbol{z}_c - \min_c}{\max_c - \min_c}, \tag{3}$$

where $\min_c$ and $\max_c$ are the minimum and maximum values for the $c$-th channel, respectively.

**Quadratic Modulation.** Next, we feed the normalized feature map into the function:

$$f_m\big(\widetilde{\boldsymbol{z}}_c; \alpha\big) = \widetilde{\boldsymbol{z}}_c + \alpha\,\widetilde{\boldsymbol{z}}_c\big(1 - \widetilde{\boldsymbol{z}}_c\big), \qquad \alpha \in (-1, 1) \tag{4}$$

Here, $\alpha$ controls both the strength and direction of modulation.

This function preserves the monotonic ordering of feature values. As illustrated in the bottom-right curve of Fig. 3, quadratic FM systematically reshapes the feature distribution: positive $\alpha$ values enhance latent activations, while negative $\alpha$ values suppress them, which can potentially help pull back out-of-distribution potential outlier features.

**Lemma.** *Let $M_c = \mathbb{E}[\widetilde{\boldsymbol{z}}_c]$ and $V_c = \text{Var}(\widetilde{\boldsymbol{z}}_c)$ denote the mean and variance of channel $c$. Under the quadratic modulation defined in Eq. 4, i) the channel mean shifts in the same direction as $\alpha$.*

*Specifically, $\alpha > 0$ increases the mean, while $\alpha < 0$ decreases the mean. ii) The channel variance is approximately rescaled by a slope-dependent factor $(1 + \alpha - 2\alpha M_c)^2$, where $\alpha > 0$, channels with large $M_c$ (close to 1) exhibit reduced variance; if $\alpha < 0$, channels with high $M_c$ experience variance.*

Please see Appendix for Proof. For $\alpha > 0$, channels with larger means receive a positive mean boost and typically show reduced variance, which strengthens high-magnitude latent responses. Reflecting on the pixel level, we observe that this often bumps up the object salience that often correspond to the main objects in the prompt, improves global cohesion, suppresses background noise and helps smooth the overall image. For $\alpha < 0$, the variance tends to increase and the dominance of high-mean channels is reduced, which allows lower-magnitude responses to surface. In images, we observe that this often recovers fine details and local textures that might otherwise be treated as noise during denoising, producing crisper micro-contrast and richer small-scale content. A further direct visualization and ablation study on $alpha$ can be found in Sec. 4.5.

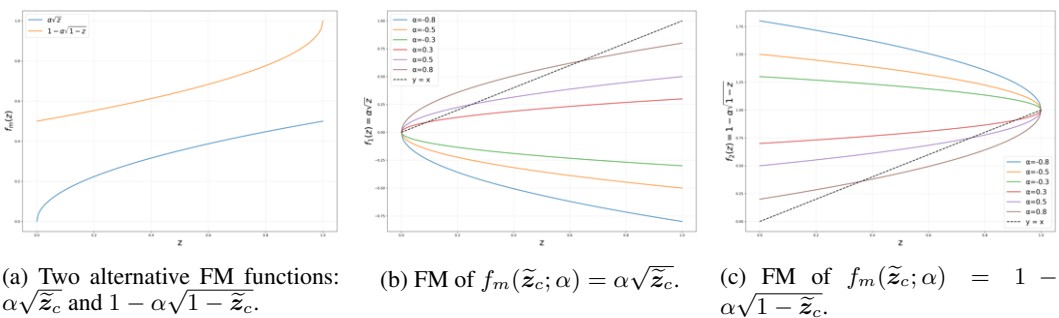

(a) Two alternative FM functions: $\alpha\sqrt{\widetilde{z}_c}$ and $1 - \alpha\sqrt{1 - \widetilde{z}_c}$.

(b) FM of $f_m(\widetilde{z}_c; \alpha) = \alpha\sqrt{\widetilde{z}_c}$.

(c) FM of $f_m(\widetilde{z}_c; \alpha) = 1 - \alpha\sqrt{1 - \widetilde{z}_c}$.

Figure 4: Alternative feature modulating functions compared against the proposed quadratic FM (Fig. 3(c)).

While alternative formulations such as $f_m(\widetilde{z}_c; \alpha) = \alpha\sqrt{\widetilde{z}_c}$ or $f_m(\widetilde{z}_c; \alpha) = 1 - \alpha\sqrt{1 - \widetilde{z}_c}$ are conceivable, they fail to preserve the original range of $\widetilde{z}_c$. Moreover, these mappings introduce asymmetric distortions to the distribution. The function curves are illustrated in Fig. 4. Other functional forms could also be employed, but we present the quadratic variant as a representative example due to its balance of simplicity and monotonicity.

**Rescaling.** To restore the original feature scale, we remap the modulated feature maps by:

$$\hat{z}_c = f_m\big(\widetilde{z}_c; \alpha\big)(\max_c - \min_c) + \min_c. \tag{5}$$

The FM function $f_M(\cdot)$ can then be defined by Eq. 3, 4 and Eq. 5.

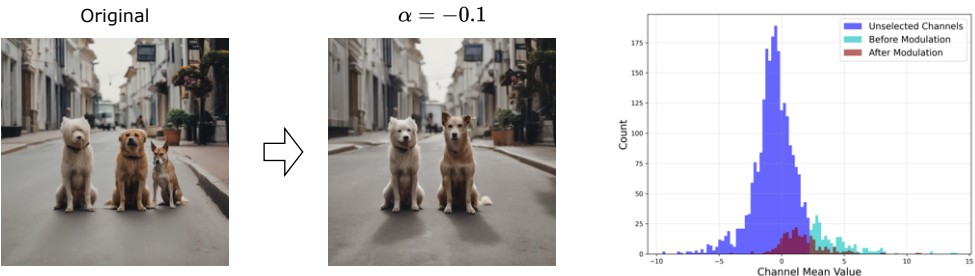

Figure 5: Feature distribution before and after feature modulation. **Left:** SDXL outputs for prompt *"Two dogs on the street"* before (left) and after FM (right). **Right:** Feature distribution shift with $\alpha = -0.1$ and channel-selection ratio $\gamma = 0.1$. Blue bars indicate unmodified channels, cyan bars represent the top 10% selected channel, and red bars depict those channels after modulation.

### 3.2.2 FEATURE MODULATION ON SELECTED CHANNELS

Applying FM indiscriminately to all channels in the latent feature map is likely to disrupt semantic consistency and collapse the denoising process. Considering the conclusion on the mean and

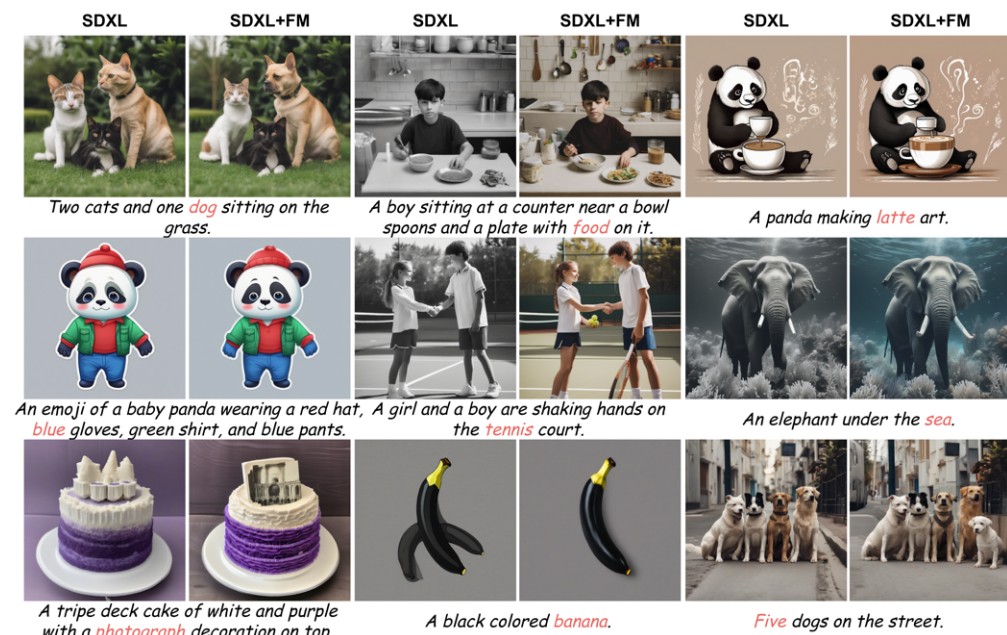

Figure 6: Comparison of SDXL outputs without and with FM. Corresponding prompts are shown below each example. The difference observed is related to words highlighted in red.

variance shifts induced by the FM function, we adopt a selective strategy: only a fraction $\gamma$ of the channels are modulated, focusing on those with the highest mean activations. The average activation of each channel is computed along the spatial dimensions as $\bar{z} = \frac{1}{H \times W} \sum_{h=1}^{H} \sum_{w=1}^{W} z$. Next, the channels are ranked in descending order of $\bar{z}$. We then select a subset $S$ corresponding to the top-$\gamma$ proportion of channels and apply modulation only to this set:

$$
\hat{z} = \begin{cases} f_M(z_c, \alpha), & \text{if } c \in S, \\ z, & \text{otherwise.} \end{cases}
\tag{6}
$$

By restricting modulation to $S$, we prevent large-scale changes in the generated images. Fig. 5 illustrates the feature distribution shift of FM with $\alpha = -0.1$: the original output incorrectly shows three dogs, while the modulated features lead to two dogs, consistent with the prompt. The histogram highlights that FM primarily reduces the mean of the selected high-activation channels (cyan → red), while leaving others unchanged (blue).

## 4 EXPERIMENTS

### 4.1 IMPLEMENTATION DETAILSEXPERIMENTAL SETUP

We evaluate the effectiveness of our FM across several versions of Stable Diffusion, including Stable Diffusion 2.1 (SD2.1) (Rombach et al., 2022), Stable Diffusion XL (SDXL) (Podell et al., 2023), and Stable Diffusion 3.0 (SD3.0) (Esser et al., 2024). These models span diverse architectures, ranging from U-Net-based designs (SD2.1 and SDXL) to the DiT-based SD3.0, demonstrating the seamless applicability of our method across a wide range of Stable Diffusion backbones.

**FM on SD2.1 and SDXL.** For U-Net based SD2.1 and SDXL, FM can be applied to features from both the skip connections and the decoder layers. While the FM can simultaneously modulate feature maps from both components across multiple layers with different $\alpha$, we simplify our analysis by adjusting a single $\alpha$ at one layer while keeping the others zero. For SD2.1 and SDXL, the channel selection ratio is set to $\gamma = 0.1$.

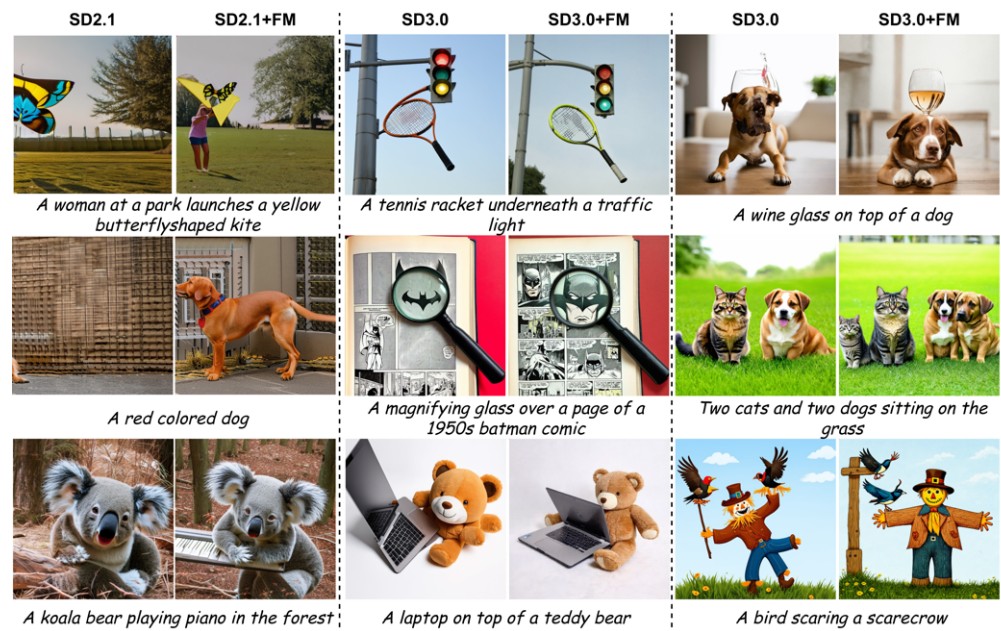

Figure 7: Visualization of outputs from SD2.1 (Rombach et al., 2022) and SD3.0 (Esser et al., 2024) without and with FM.

**FM on SD3.0.** SD3.0 is based on DiT, a pure Transformer-based architecture designed for diffusion models. We observe that DiT models are sensitive to channel-level feature modifications. To ensure the stability of DiT with FM, we apply it to the features in the final MM-DiT block.

Positioned near the output stage, this block offers two key advantages: first, it enables FM to effectively influence the final generation; second, modifications at this stage have a localized effect that minimizes disruption to earlier layers, thereby preserving the overall stability of the network. For SD3.0, the channel selection ratio is set to $\gamma = 0.5$.

## 4.2 VISUALIZATION ANALYSIS

We present qualitative results in Figs. 6 and 7, illustrating image pairs generated by the three aforementioned Stable Diffusion models before and after applying FM, given various prompts. As observed, FM provides promising improvements across these examples. Based on our observations, we categorize the improvements into four distinct types.

**Integrity Improvement.** A common problem in generative models is the generation of corrupted or incomplete outputs. FM can adjust feature maps to effectively correct such corruptions. For instance, in one of the subfigures in Fig. 6, SDXL generates a defective image of a banana when given the prompt *"A black colored banana."*. By modulating the feature maps, FM rectifies this flawed generation, producing a semantically accurate image of the black banana. A similar phenomenon is observed in Fig. 7, where FM effectively fixes the severely corrupted outputs generated by SD3.0 (e.g., *"A laptop on top of a teddy bear."* and *"A wine glass on top of a dog."*).

**Improving Numerical Accuracy.** Another challenge for diffusion models is accurately interpreting numerical specifications in prompts. As shown in Figs. 6 and 7, Stable Diffusion sometimes misrepresents the number of subjects, demonstrated by prompts such as *"Two cats and one dog sitting on the grass."* and *"Five dogs on the street."* in Fig. 6, and *"Two cats and two dogs sitting on the grass."* in Figs. 7. FM helps the model achieve better alignment with the prompt in terms of specified object counts.

**Completeness.** Diffusion models sometimes fails to incorporate certain objects or details into the generated image, especially when the prompt is complex. In Fig. 6, for instance, the left of the second image pair in the first row lacks the *"food"* mentioned in the text prompt. Applying FM recovers the missing food in the final output. Likewise, the prompt *"A panda making latte art"*

regains its *"latte art"* detail in the generated image. Missing element *"photograph"* are restored for the prompt of the cake. Fig. 7 shows analogous improvements in SD2.1, where absent objects like a *"woman"*, a *"red-colored dog"*, and a *"piano"* are reintroduced in the images.

**Visual Refinement.** FM have also shown an ability on refine colorization and image quality. Visual refinement is subjective to individual users. In Fig. 6, FM yields more vibrant colors and sharper details for the *"elephant under the sea"*.

## 4.3 EVALUATION ON LLM

Due to the fact that the optimal setting of $\alpha$ is highly image-dependent and varies significantly across different cases, conducting a fair statistical evaluation (such as CLIP score, please see the Appendix) becomes challenging. Since the optimal setting of $\alpha$ may vary from image to image, even though multiple $\alpha$ values can improve a given example, conducting a strictly fair statistical evaluation (such as CLIP score, please see the Appendix) remains non-trivial. This is primarily because searching for the optimal $\alpha$ incurs considerable computational overhead. To address this, we conduct a rigorous evaluation using a large language model (LLM), specifically GPT-4o.

**Dataset Prepare.** To validate the reliability and discriminative capability of the LLM in distinguishing image quality before and after FM, we manually constructed an annotated dataset. This dataset consists of $47$ positive sample pairs indicating improvement after FM, and $30$ negative sample pairs indicating no improvement or degradation. Among the negative samples, $20$ represent ambiguous cases with negligible visual difference, while the remaining $10$ clearly exhibit degraded quality.

Table 1: Evaluation results on our annotated datasets. The current prompt design achieves an overall accuracy of $88.31\%$.

| | **Positive** | | | **Negative** | | |
|---|---|---|---|---|---|---|
| Total | 47 | | | 30 | | |
| Criterion | *Yes* | *No* | *Similar* | *Yes* | *No* | *Similar* |
| Integrity | 24 (51.1%) | 1 (2.1%) | 22 (46.8%) | 2 (6.7%) | 4 (13.3%) | 24 (80.0%) |
| Numerical | 10 (21.3%) | 1 (2.1%) | 36 (76.6%) | 0 (0.0%) | 0 (0.0%) | 30 (100.0%) |
| Completeness | 27 (57.4%) | 1 (2.1%) | 19 (40.4%) | 1 (3.3%) | 2 (6.7%) | 27 (90.0%) |
| Visual Ref. | 24 (51.1%) | 1 (2.1%) | 22 (46.8%) | 2 (6.7%) | 6 (20.0%) | 22 (73.3%) |
| **Overall** | 40 (85.1%) | 1 (2.1%) | 6 (12.8%) | 2 (6.7%) | 7 (23.3%) | 21 (70.0%) |
| **Accuracy** | **40 (85.1%)** | | | **28 (93.33%)** | | |

**Prompt Design.** We design the evaluation prompt for LLM based on our visual observations. For each image pair (Original vs. FM-processed), the LLM is asked to return judgement for each of four criteria: 1) **Integrity:** Whether the FM-processed image helps recovering corruptions. 2) **Numerical Accuracy:** Whether the FM-processed image shows any improvement in numeric values based on the text prompt. 3) **Completeness:** Whether the FM brings back any missing objects. 4) **Visual Refinement:** Whether FM improves color consistency, contrast, sharpness, or noise level.

**Final Judgment.** Each criterion has three answers: "Yes", "No" and "Similar". The final judgment for each prompt is determined as follows: **"No"**: if *any* criterion is "No,"; **"Yes"**: if *no* criterion is "No" and *at least one* is "Yes,". **"Similar"**: if *all* criteria are "Similar."

**Evaluation on Annotated Dataset.** We first tested our evaluation prompt with GPT-4o on our labeled dataset to ensure the LLM has the required ability for further evaluation. The results presented in Tab. 1 demonstrate the strong alignment between LLM judgments and human annotations.

**Evaluation on Benchmark Dataset.** We extend our evaluation to the DRAWBENCH, COCO, and a small customized prompt set using the exact same GPT-4o prompt. For the choice of optimal $\alpha$ for each prompt, we implement a simple grid search with $\alpha \in \{-0.2, 0.2\}$. We apply FM with $\alpha = -0.2$, and if no improvement is found, continue to $\alpha = 0.2$. If there exists an $\alpha$ that returns "Yes" as the final judgment, we stop and move to the next prompt. As shown in Tab. 2, FM achieved an overall improvement rate of $57.3\%$, while the majority of the remaining cases remain "Similar." Recognizing that $\{-0.2, 0.2\}$ may not capture the optimal modulation, we expand $\alpha$ to $\alpha_m \in \{-0.2, -0.1, 0.1, 0.2\}$, increasing the overall improvement rate to $72.7\%$.

Table 2: Evaluaion results on GPT-4o on benchmark prompts. Percentages are relative to the total of the three outcome counts (*Yes / No / Similar*) for each criterion within each dataset and overall. The last column stands for the improvement rate.

| | DRAWBENCH | | | COCO | | | Custom | | | Impr. Rate |
|---|---|---|---|---|---|---|---|---|---|---|
| Total | 200 | | | 999 | | | 29 | | | 1229 |
| Criterion | *Yes* | *No* | *Similar* | *Yes* | *No* | *Similar* | *Yes* | *No* | *Similar* | *Yes* |
| | | | | | $\alpha \in \{-0.2, 0.2\}$ | | | | | | |
| Integrity | 62 (31.0%) | 0 (0.0%) | 138 (69.0%) | 305 (30.6%) | 16 (1.6%) | 677 (67.8%) | 6 (20.7%) | 0 (0.0%) | 23 (79.3%) | 30.3% |
| Numerical | 21 (10.5%) | 1 (0.5%) | 178 (89.0%) | 57 (5.7%) | 1 (0.1%) | 940 (94.2%) | 0 (0.0%) | 0 (0.0%) | 29 (100.0%) | 6.3% |
| Completeness | 81 (40.5%) | 1 (0.5%) | 118 (59.0%) | 273 (27.4%) | 5 (0.5%) | 720 (72.1%) | 11 (37.9%) | 0 (0.0%) | 18 (62.1%) | 29.7% |
| Visual Ref. | 124 (62.0%) | 1 (0.5%) | 75 (37.5%) | 427 (42.8%) | 12 (1.2%) | 559 (56.0%) | 16 (55.2%) | 0 (0.0%) | 13 (44.8%) | 46.1% |
| **Overall** | **153 (76.5%)** | **2 (1.0%)** | **45 (22.5%)** | **531 (53.2%)** | **21 (2.1%)** | **446 (44.7%)** | **20 (69.0%)** | **0 (0.0%)** | **9 (31.0%)** | **57.3%** |
| | | | | | $\alpha \in \{-0.2, -0.1, 0.1, 0.2\}$ | | | | | | |
| Integrity | 75 (37.5%) | 0 (0.0%) | 125 (62.5%) | 368 (36.8%) | 5 (0.5%) | 626 (62.7%) | 11 (37.9%) | 0 (0.0%) | 18 (62.1%) | 37.0% |
| Numerical | 21 (10.5%) | 0 (0.0%) | 179 (89.5%) | 62 (6.2%) | 2 (0.2%) | 935 (93.6%) | 2 (6.9%) | 0 (0.0%) | 27 (93.1%) | 6.9% |
| Completeness | 91 (45.5%) | 0 (0.0%) | 109 (54.5%) | 351 (35.1%) | 5 (0.5%) | 643 (64.4%) | 13 (44.8%) | 0 (0.0%) | 16 (55.2%) | 37.1% |
| Visual Ref. | 123 (61.5%) | 1 (0.5%) | 76 (38.0%) | 536 (53.7%) | 12 (1.2%) | 451 (45.1%) | 16 (55.2%) | 0 (0.0%) | 13 (44.8%) | 54.9% |
| **Overall** | **167 (83.5%)** | **1 (0.5%)** | **32 (16.0%)** | **702 (70.3%)** | **16 (1.6%)** | **281 (28.1%)** | **24 (82.8%)** | **0 (0.0%)** | **5 (17.2%)** | **72.7%** |

Table 3: Distribution of the first $\alpha$ that yields an improvement under GPT-4o for each dataset and search setting (two candidates vs. four). Most improved prompts are fixed by $\alpha = -0.2$ or $\alpha = -0.1$, so only one or two FM samples beyond the baseline are typically required.

| Dataset | $\alpha$ set | Improved | $-0.2$ | $-0.1$ | 0.1 | 0.2 |
|---|---|---|---|---|---|---|
| Custom | $\{-0.2, 0.2\}$ | 20 | 17 (85.0%) | – | – | 3 (15.0%) |
| | $\{-0.2, -0.1, 0.1, 0.2\}$ | 24 | 15 (62.5%) | 6 (25.0%) | 2 ( 8.3%) | 1 ( 4.2%) |
| DrawBench | $\{-0.2, 0.2\}$ | 153 | 130 (85.0%) | – | – | 23 (15.0%) |
| | $\{-0.2, -0.1, 0.1, 0.2\}$ | 167 | 116 (69.5%) | 26 (15.6%) | 15 ( 9.0%) | 10 ( 6.0%) |
| COCO | $\{-0.2, 0.2\}$ | 531 | 355 (66.9%) | – | – | 176 (33.1%) |
| | $\{-0.2, -0.1, 0.1, 0.2\}$ | 702 | 367 (52.3%) | 156 (22.2%) | 128 (18.2%) | 51 ( 7.3%) |

## 4.4 EVALUATION ON T2I-COMPBENCH

We further evaluate FM on T2I-CompBench++(Huang et al., 2023; 5555), a compositional text-to-image benchmark. For a fair comparison with existing training-free methods, we follow the official T2I-CompBench++ evaluation code and default hyperparameters, and restrict all methods to the two backbones supported by the benchmark, SD2.1 and SDXL. The benchmark only provides official implementations for FreeU and Attend-and-Excite on these backbones; we therefore evaluate FM under exactly the same settings. For FM on SD2.1 and SDXL, we adopt the same per-image search strategy as in our GPT-4o evaluation, selecting the best-performing $\alpha$ for each prompt from the fixed grid $\{-0.2, -0.1, 0.1, 0.2\}$. The detailed compositional results are reported in Tab. 4.

Table 4: T2I-CompBench++ results for baseline models, FreeU, Attend-and-Excite, and FM on SD2.1 and SDXL. All methods use the official benchmark code and default hyperparameters.

| Method | Backbone | Color | Shape | Texture | Numeracy | 3D | Non-spat. | Spat. |
|---|---|---|---|---|---|---|---|---|
| Baseline | SD2.1 | 0.40±0.35 | 0.39±0.30 | 0.47±0.32 | 0.34±0.23 | 0.23±0.16 | 0.30±0.03 | 0.05±0.21 |
| | SDXL | 0.60±0.36 | 0.48±0.31 | 0.54±0.37 | 0.49±0.24 | 0.37±0.18 | 0.31±0.03 | 0.23±0.40 |
| FreeU | SD2.1 | 0.58±0.14 | 0.57±0.14 | 0.55±0.15 | 0.46±0.27 | 0.34±0.17 | 0.31±0.03 | 0.19±0.37 |
| | SDXL | 0.57±0.15 | 0.52±0.15 | 0.51±0.14 | 0.44±0.25 | 0.31±0.03 | 0.31±0.03 | 0.22±0.39 |
| Attend-and-Excite | SD2.1 | 0.49±0.15 | 0.48±0.15 | 0.47±0.15 | 0.45±0.25 | 0.33±0.18 | 0.31±0.03 | 0.18±0.36 |
| FM | SD2.1 | 0.58±0.33 | 0.58±0.30 | 0.58±0.32 | 0.49±0.23 | 0.37±0.17 | 0.31±0.03 | 0.21±0.30 |
| | SDXL | **0.73±0.30** | **0.62±0.30** | **0.68±0.34** | **0.63±0.25** | **0.48±0.20** | **0.32±0.03** | **0.31±0.44** |

Table 5 summarizes which $\alpha \in \{-0.2, -0.1, 0.1, 0.2\}$ is selected as the per-image best value for SD2.1 and SDXL.

## 4.5 ABLATION STUDY ON IMPACT OF $\alpha$.

Table 5: Per-image best $\alpha$ distribution on T2I-CompBench++ (percentage of images).

| | SD2.1 | | | | SDXL | | | |
|---|---|---|---|---|---|---|---|---|
| Dataset | $-0.2$ | $-0.1$ | 0.1 | 0.2 | $-0.2$ | $-0.1$ | 0.1 | 0.2 |
| color | 31.3 | 22.3 | 19.0 | 27.3 | 39.0 | 14.3 | 15.3 | 31.3 |
| shape | 25.0 | 24.3 | 24.7 | 26.0 | 33.7 | 20.3 | 18.3 | 27.7 |
| texture | 34.0 | 22.3 | 25.0 | 18.7 | 31.3 | 17.7 | 19.3 | 31.7 |
| spatial | 91.3 | 3.3 | 0.7 | 4.7 | 79.0 | 7.3 | 7.0 | 6.7 |
| 3D spatial | 37.0 | 25.0 | 18.0 | 20.0 | 41.0 | 18.0 | 13.7 | 27.3 |
| numeracy | 69.7 | 12.3 | 11.3 | 6.7 | 58.0 | 18.0 | 12.7 | 11.3 |
| non-spatial | 29.3 | 24.0 | 23.0 | 23.7 | 32.7 | 18.0 | 14.0 | 35.3 |
| overall | 45.4 | 19.1 | 17.4 | 18.1 | 44.9 | 16.2 | 14.3 | 24.5 |

Fig. 8 Figure 8 illustrates how varying the value of $\alpha$ affects the output images across three representative examples. The optimal $\alpha$ is prompt dependent. In the first case, increasing $\alpha$ improves the accuracy of the dog count in the image. In the second example, a small negative $\alpha$ effectively reduces feature intensity, making the "person" more visible. These observations are consistent with the inference in Sec. 3.2, where a positive $\alpha$ tends to enhance the primary objects in the prompt and produce smoother images, while a negative $\alpha$ sharpens local details and can bring back out-of-distribution potential outlier features. Please see Appendix for further ablation study.

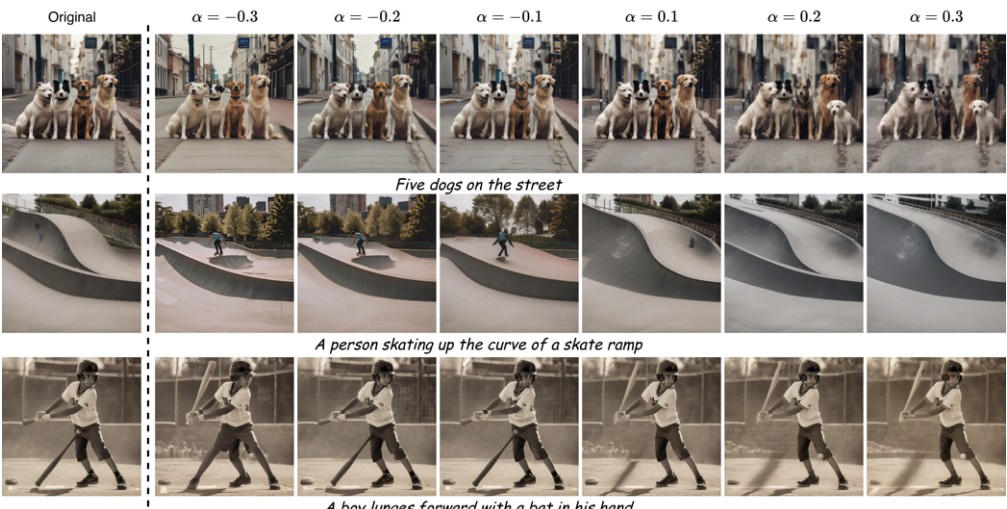

Figure 8: Ablation study on the effect of different values of $\alpha$. We present results for three prompts. The original images (left) serve as reference, and a channel-selection ratio $\gamma = 0.1$ is used.

Based on our experiments, the optimal value of $\alpha$ typically lies within the range $[-0.2, 0.2]$, as values with larger magnitudes often result in noticeable image degradation.

## 5 CONCLUSIONS AND FUTURE WORK

In this study, we introduced FM, a training-free method that significantly enhances the sample quality of diffusion models without incurring any additional computational costs without introducing any additional training cost. By modulating latent feature maps during the denoising steps, FM improves text-image alignment, overall image quality, and effectively corrects artifacts such as broken outputs. Our experiments across various Stable Diffusion models have demonstrated clear improvements, highlighting the potential of selective channel-level modulation as a lightweight yet powerful tool for enhancing generative performance.

**Future Work.** In future work, we plan to extend FM to other diffusion model architectures and explore its application in video generation tasks. Additionally, we aim to investigate the integration of FM as a trainable layer by making the modulating parameter $\alpha$ learnable.

## LLM USAGE STATEMENT

In this paper, LLMs were used solely for academic writing assistance, including grammar correction, improvements in fluency, and polishing of presentation style. All text generated through LLM assistance was subsequently reviewed and manually corrected to ensure accuracy and consistency. The role of LLMs was strictly limited to improving readability and formatting, without contributing to the scientific content or experimental results.

## REPRODUCIBILITY STATEMENT

We provide the core implementation and a runnable demo in the supplementary material to support full reproducibility. Owing to the nature of our proposed Feature Modulation (FM) method, each prompt–image pair may achieve its best improvement under a prompt-specific hyperparameter $\alpha$. As such, exact reproduction of every example requires tuning $\alpha$ on a per-case basis. Nevertheless, the provided code and demo instructions enable faithful reproduction of the main results, and we include several illustrative examples corresponding to those presented in the paper.

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

# APPENDIX OF FEATURE MODULATING FOR DIFFUSION MODELS

## A  PROMPTS FOR FIGURE 1

Below are the prompts used to generate the images in Fig. 1:

- Top Left Group: *A boy lunges forward with a bat in his hand.*
- Middle Left Group: *Two dogs on the street.*
- Top Right Group: *A person skating up the curve of a skate ramp.*
- Bottom Left Group: *A mascot with people on a baseball field.*
- Bottom Middle Group: *A black colored dog.*
- Bottom Right Group: *A triangular purple flower pot. A purple flower pot in the shape of a triangle.*

## B  DISTRIBUTION CHARACTERISTIC OF FM

Here we provide the proof of **Lemma** introduced in Sec. 3.2.

**Lemma.** *Let $M_c = \mathbb{E}[\widetilde{z}_c]$ and $V_c = \mathrm{Var}(\widetilde{z}_c)$ denote the mean and variance of channel c. Under the quadratic modulation defined in Eq. 4, i) the channel mean shifts in the same direction as $\alpha$. Specifically, $\alpha > 0$ increases the mean, while $\alpha < 0$ decreases the mean. ii) The channel variance is approximately rescaled by a slope-dependent factor $(1 + \alpha - 2\alpha M_c)^2$, where $\alpha > 0$, channels with large $M_c$ (close to 1) exhibit reduced variance; if $\alpha < 0$, channels with high $M_c$ experience variance. Please see Appendix for Proof.*

**Proof. i) Mean Sift.** From Eq. 4, the post-modulation mean is

$$M_c' = \mathbb{E}[f_m(\widetilde{z}_c; \alpha)] \tag{7}$$

$$= (1 + \alpha)M_c - \alpha\,\mathbb{E}[\widetilde{z}_c^2] \tag{8}$$

$$= (1 + \alpha)M_c - \alpha(M_c^2 + V_c). \tag{9}$$

Hence the mean shift is

$$\Delta M_c = M_c' - M_c = \alpha\,(M_c - M_c^2 - V_c). \tag{10}$$

Since $\widetilde{z}_c \in (0, 1]$, it follows that $\mathbb{E}[\widetilde{z}_c^2] \leq \mathbb{E}[\widetilde{z}_c] = M_c$, which implies $V_c \leq M_c - M_c^2$. Consequently, $M_c - M_c^2 - V_c \geq 0$. Therefore, the channel mean shifts in the same direction as $\alpha$.

**ii) Variance Sift.** The variance after modulation is

$$\mathrm{Var}[f_m(\widetilde{z}_c; \alpha)] = \mathbb{E}[f_m(\widetilde{z}_c; \alpha)^2] - (M_c')^2. \tag{11}$$

This expression is exact and depends on higher-order moments of $\widetilde{z}_c$, such as $\mathbb{E}[\widetilde{z}_c^3]$ and $\mathbb{E}[\widetilde{z}_c^4]$. To obtain a tractable form that characterizes the leading-order effect of $\alpha$, we employ a first–order delta method expansion around $M_c$. This is justified under the assumptions established in Sec. 3.2: $\alpha \in [-1, 1]$ and $\widetilde{z}_c \in (0, 1]$ due to normalization. Under these conditions, the modulation $f_m$ remains smooth and bounded on the support, ensuring that higher-order terms of the Taylor expansion contribute only limited corrections. The delta method gives

$$\mathrm{Var}[g(\widetilde{z}_c)] \;\approx\; (g'(M_c))^2\,V_c,$$

with $g'(x) = 1 + \alpha - 2\alpha x$. Substituting gives

$$\mathrm{Var}[f_m(\widetilde{z}_c; \alpha)] \;\approx\; (1 + \alpha - 2\alpha M_c)^2\,V_c. \tag{12}$$

Thus the variance is approximately scaled by the factor $(1 + \alpha - 2\alpha M_c)^2$. In particular,

- if $\alpha > 0$, channels with large $M_c$ (close to 1) tend to exhibit reduced variance;
- if $\alpha < 0$, channels with large $M_c$ tend to undergo variance amplification.

## C    EFFECT ON THE DENOISING FIELD

To connect channel-wise FM to the global reverse process, we view the network at timestep $t$ as defining a time-dependent denoising field $F(\boldsymbol{x}_t, t)$. In our encoder–decoder decomposition,

$$F_0(\boldsymbol{x}_t, t) := f_{\text{dec}}\big(f_{\text{enc}}(\boldsymbol{x}_t, t), t\big) \tag{13}$$

denotes the original (unmodulated) field, while applying FM to the encoder features yields

$$F_\alpha(\boldsymbol{x}_t, t) := f_{\text{dec}}\big(f_M(f_{\text{enc}}(\boldsymbol{x}_t, t); \alpha), t\big), \tag{14}$$

with perturbation $\Delta F(\boldsymbol{x}_t, t; \alpha) = F_\alpha(\boldsymbol{x}_t, t) - F_0(\boldsymbol{x}_t, t)$.

Assuming $\alpha$ is of moderate magnitude, a first-order Taylor expansion around $\alpha = 0$ gives

$$F_\alpha(\boldsymbol{x}_t, t) \approx F_0(\boldsymbol{x}_t, t) + \mathcal{A}(\boldsymbol{x}_t, t)\, \alpha, \tag{15}$$

where $\mathcal{A}(\boldsymbol{x}_t, t)$ collects the decoder sensitivity to the selected top-$\gamma$ channels and the derivative of $f_M$ with respect to $\alpha$. Thus FM acts as a linear perturbation of the denoising field along dominant semantic feature directions: positive $\alpha$ biases the reverse dynamics toward strengthening high-activation channels (main objects and global structure), whereas negative $\alpha$ suppresses them and relatively emphasizes weaker channels (local textures and fine details).

## D    MULTIPLE FEATURE MODULATION

While FM is defined to operate for one feature map, it is possible to simultaneously apply FM at different layers of the diffusion model during the denoising process. This could further improve performance compared to single-layer modulation. We consider a U-Net like architecture in this case. Let $l \in \{1, 2, \ldots, L\}$ represent different feature levels. Deeper layers generally exhibit more pronounced modulation effects. When FM is applied across multiple layers, separate modulation parameters are utilized. For instance, given feature level $l$, we first obtain feature maps as $\boldsymbol{z}_l = f_{\text{enc}}(x_t, t; \theta_{\mathcal{E}_l})$, where $\mathcal{E}_l(\cdot)$ denotes the mapping to level $l$. These features are independently modulated to yield $\hat{\boldsymbol{z}}_l = f_M(\boldsymbol{z}_l, t; \alpha_l)$, and the resulting latent representation for the subsequent diffusion step is computed as $\hat{\epsilon} = f_{\text{dec}}\big(\hat{\boldsymbol{z}}_0, \hat{\boldsymbol{z}}_1, \ldots, \hat{\boldsymbol{z}}_L, t; \theta_{\mathcal{D}}\big)..$

## E    VISUALIZATIONS OF THE FM EFFECT

We visualize how the quadratic mapping $f_m(\tilde{z}; \alpha) = \tilde{z} + \alpha\, \tilde{z}(1 - \tilde{z})$ reshapes a normalized channel while keeping value ordering intact. We sample one top-$\gamma$ channel from the SDXL skip feature at a mid denoising step ($t{=}30$), normalize it to $(0, 1]$, and compare distributions before and after applying FM.

Across both cases, the curves never cross (ordering preserved) while the empirical mean/variance move in the directions predicted by the lemma, providing an intuitive distributional illustration of FM's modulation effect.

## F    DIFFERENT FM FOR DIFFERENT STEPS

The FM modulation does not need to use a single strength at all timesteps. In principle, one can apply FM only to a subset of denoising steps or use a timestep-dependent schedule. Fig. F11 compares several such choices, including variants that restrict FM to later steps, and they all still produce clear improvements over the baseline. For simplicity and consistency, all main experiments in the paper use one global $\alpha$ shared by all denoising timesteps, which we find sufficient to obtain strong gains.

## G    EXPERIMENTS ON DIFFERENT RESOLUTIONS

We also conduct experiments on different image resolutions. For each example in Fig. G12 and Fig. G13, we generate images at four resolutions (512×512, 768×768, 1024×1024 as the default resolution, and 1280×1280) and visualize results for $\alpha$ values ranging from $-0.2$ to $0.2$.

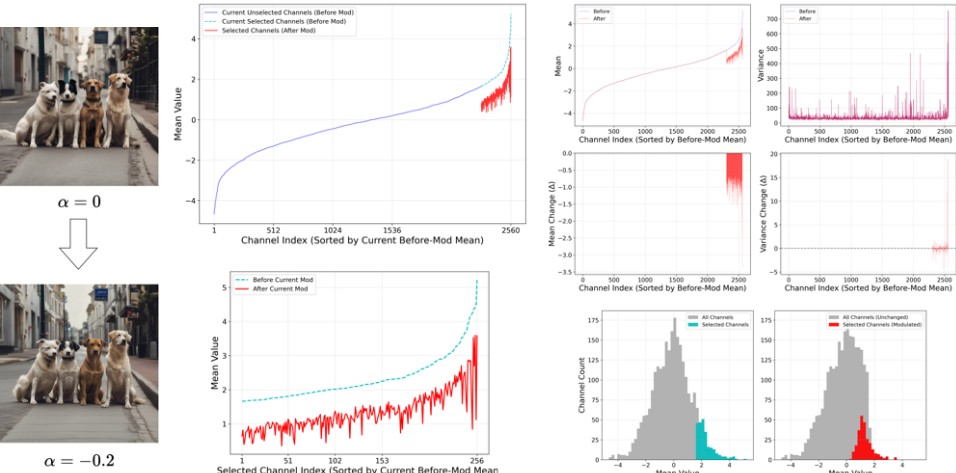

Figure E9: Simulation example with $\alpha$ decreasing from $0$ to $-0.2$. The quadratic curve (left) remains monotonic, the histogram of the selected channel shifts its mean leftward with a slight variance expansion (middle), and the downstream feature response illustrates how negative $\alpha$ suppresses dominant activations while revealing weaker structures (right), consistent with the lemma.

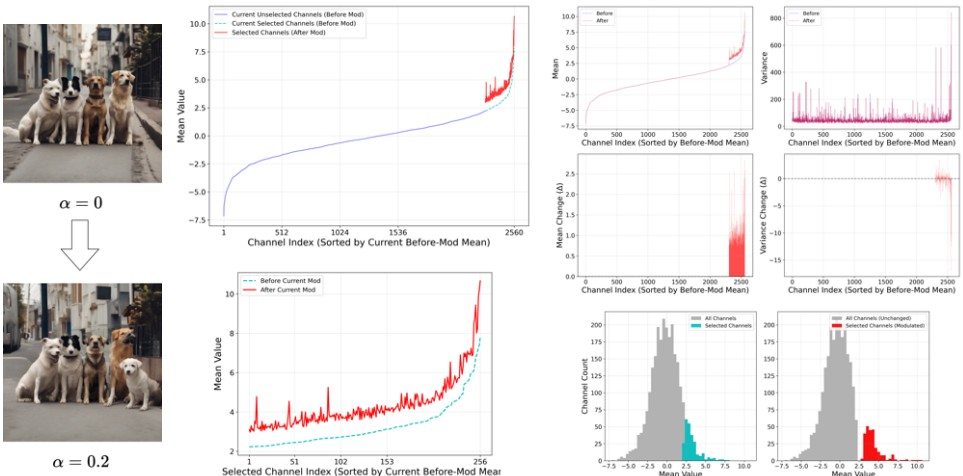

Figure E10: Simulation example with $\alpha$ increasing from $0$ to $0.2$. The mapping again preserves ordering but pushes mass toward larger values, raising the channel mean and slightly contracting the variance. The right panel shows stronger activation on the same semantic region, aligning with the theoretical monotone mean shift in the lemma.

From these figures we observe that FM is effective across different resolutions, but additional tuning is needed to obtain the best results.

We clarify that optimal setting of $\alpha$ is indeed image-dependent, varying across different cases. Fig. K18 shows that even with a fixed $\alpha$—rather than per-image optimal values—FM yields consistent, marginal improvements rather than performance degradation across the whole benchmark. This actually demonstrates FM's robustness: as a training-free method, it can benefit even "blind" cases without parameter tuning.

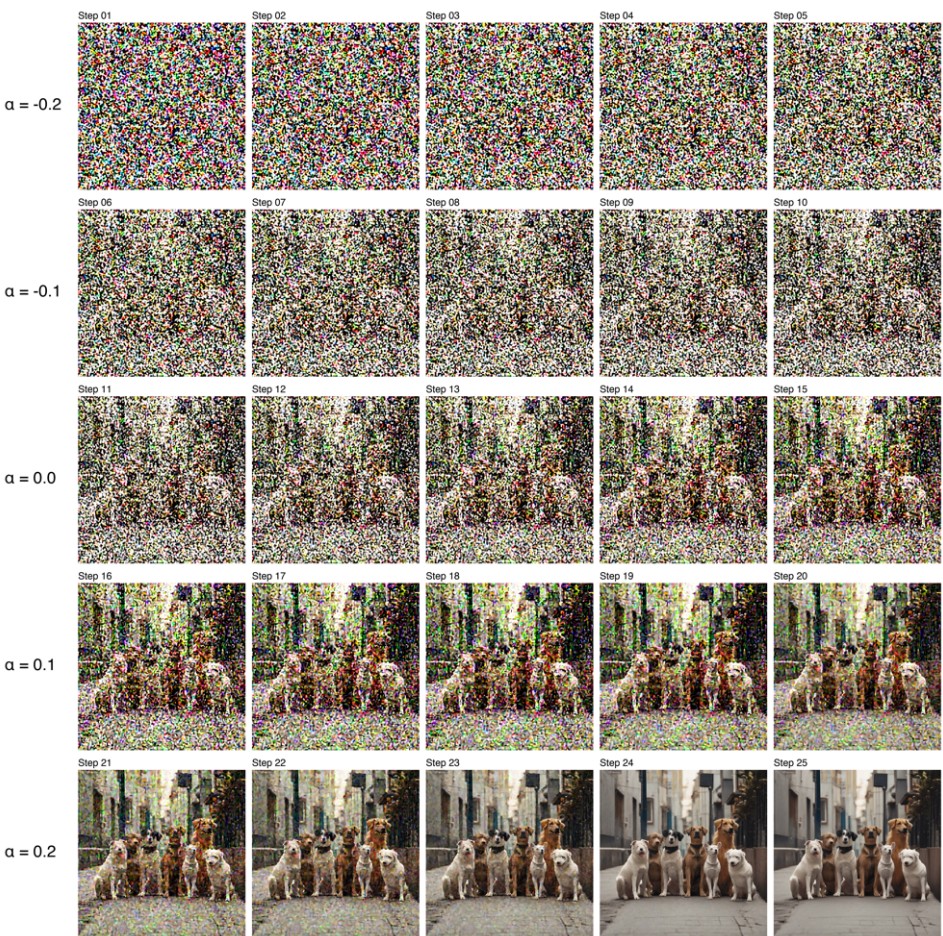

Figure F11: Effect of applying FM at different denoising steps. We use a 25-step sampler and compare five ways of distributing the modulation strength $\alpha$ across timesteps.

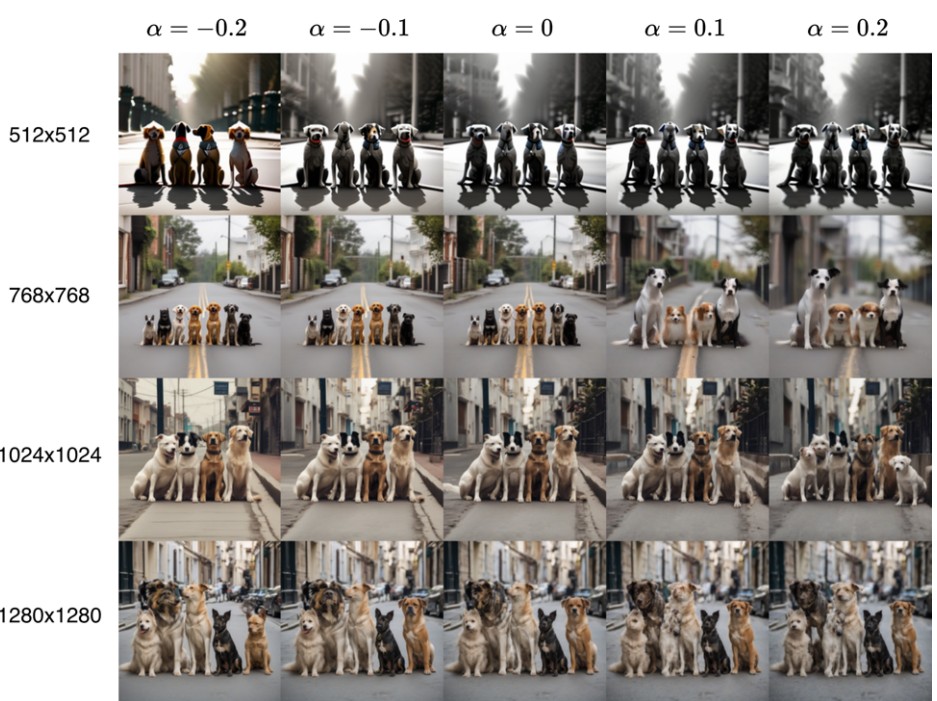

Figure G12: Different resolutions on SDXL. Prompt: *Five dogs on the street*.

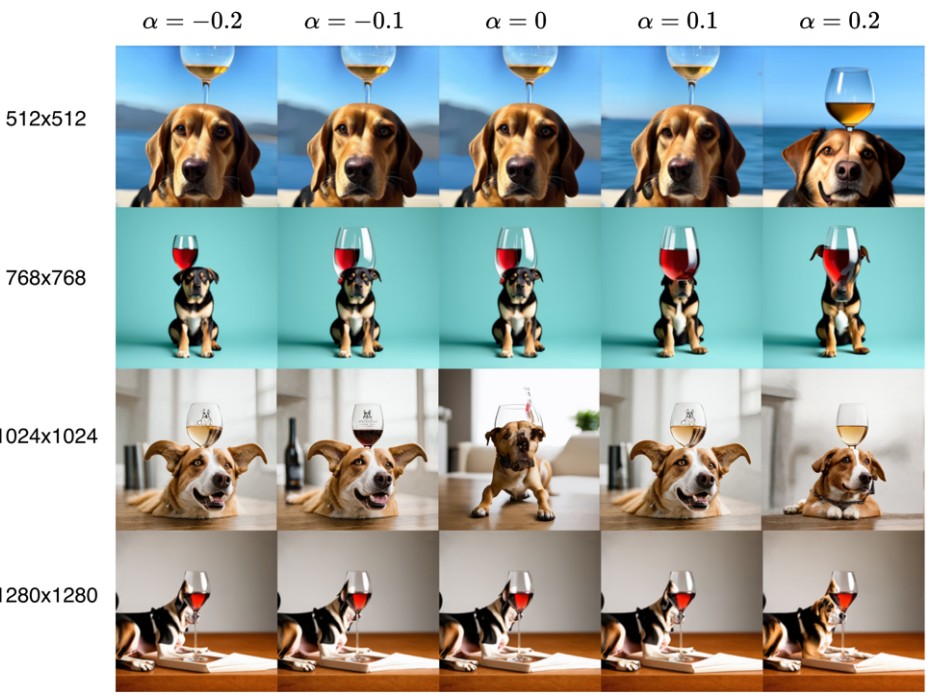

Figure G13: Different resolutions on SD3. Prompt: *A wine glass on top of a dog*.

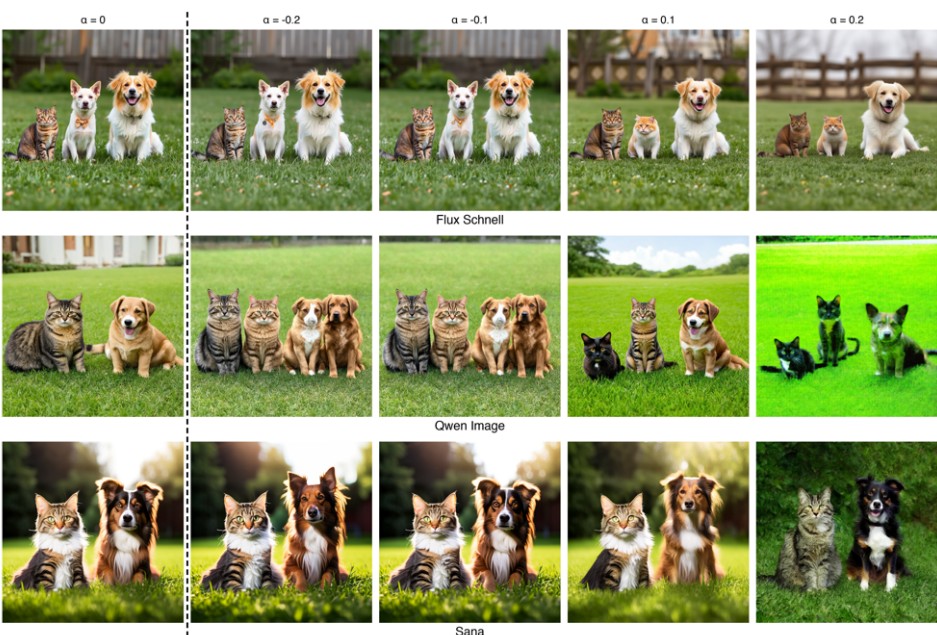

Figure H14: Prompt: *Two cats and two dogs sitting on the grass.*

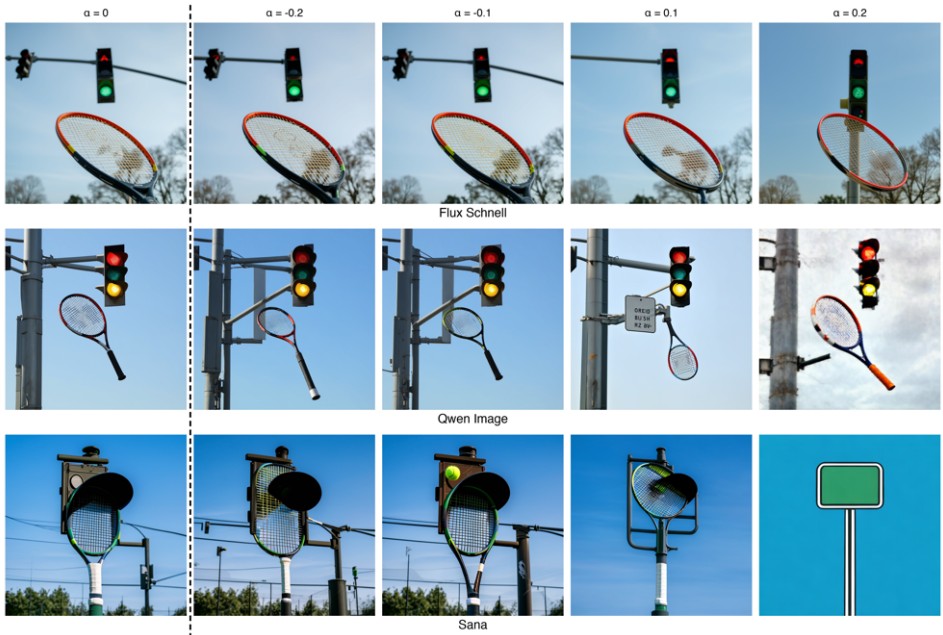

Figure H15: Prompt: *A tennis racket underneath a traffic light.*

## H  FM ON MORE MODEL ARCHITECTURES

To demonstrate the effectiveness of FM on different model structures, we apply FM on three more architectures: Flux, SANA, and Qwen-Image. Note that we simply adapt the same configuration as SD3 and the same prompts presented in Fig. 7 without any additional fine-tuning and parameter tuning. The channel ratio $\gamma$ is also set to $0.5$ for all architectures.

The results of Flux, SANA, and Qwen-Image are shown in Fig. H14, Fig. H15. Note that better results can be achieved with a better configuration of the model. We simply demonstrate the effec-

tiveness of FM on these architectures that can help balance the text to image alignment and visual qualities.

## I    CLARIFICATION OF THE "FIVE DOGS" EXAMPLE IN FIG. 8

We are deeply sorry for the mistake in the original Fig. 8. Here we provide the full original images for the prompt *"Five dogs on the street"* across all $\alpha$ values and FM layers for SDXL; the layer definitions follow Sec. R.

## J    VISUALIZATION COMPARIASON WITH BASELINE METHODS

In this section, we provide visual comparisons with two baseline methods, FreeU and Attend-and-Excite, evaluated on both SD2.1 and SDXL architectures. As shown in Fig. J17, we utilize the same prompts used in Fig. 8 and Fig. 7 to ensure a consistent evaluation.

## K    CLIP SCORES

Beyond the qualitative visualization results, we also conduct a quantitative study using the CLIP score (Hessel et al., 2021) on two standard benchmarks: COCO and DRAWBENCH (Chen, 2023). Specifically, we use SDXL and fix the parameter $\alpha$ for all prompts in these benchmarks, then evaluate the overall performance. Fig. K18 illustrates CLIP score results for both datasets with the FM on SDXL. From Fig. K18(a), we observe consistent improvement when using positive $\alpha_b$ values. Most of the positive $\alpha_s$ also lead to improvment of CLIP score. In Fig. K18(b), moderate $\alpha$ values in both the backbone and skip pathways lead to noticeable gains in CLIP scores. These findings suggest that even a simple, fixed selection of $\alpha$ values can systematically improve text-image alignment.

## L    EXPERIMENTAL SETTINGS

Here we summarize the diffusion settings used in our main experiments and T2I-CompBench evaluation.

- **SDXL:** Sampler/Scheduler = EulerDiscreteScheduler (Diffusers default); classifier-free guidance scale (CFG) 7.5; 25 sampling steps; image resolution $1024 \times 1024$; random seed 42 (default); empty negative prompt.
- **SD2.1:** Sampler/Scheduler = PNDMScheduler; CFG 7.5; 25 sampling steps; image resolution $512 \times 512$.
- **SD3.0:** Sampler/Scheduler = FlowMatchEulerDiscreteScheduler; CFG 7.0; 28 sampling steps; image resolution $1024 \times 1024$; FM applied only on a single modulation parameter in the final DiT block.

We clarify that optimal setting of $\alpha$ is indeed image-dependent, varying across different cases. Fig. K18 shows that even with a fixed $\alpha$—rather than per-image optimal values—FM yields consistent, marginal improvements rather than performance degradation across the whole benchmark. This actrually demonstrates FM's robustness: as a training-free method, it can benefit even "blind" cases without parameter tuning.

## M    RESULTS ON DIFFERENT SEEDS AND SAMPLERS

We further investigate the robustness of FM under different random seeds and samplers. In Fig. M19 we plot results on SDXL with the default sampling settings summarized in Appendix L, comparing two random seeds (42 and 300) and two other samplers: euler and DDIM. Fig. M19 shows that different samplers have small impact on the FM effect. Changing the seed alters the entire generated image and therefore affects the modulation process. As illustrated by the seed 300 results, there often exists alternative $\alpha$ value within our small grid that can still improve a suboptimal baseline image.

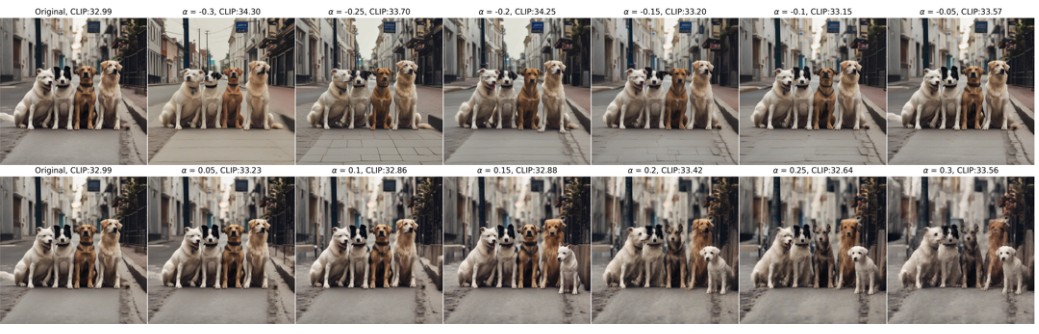

(a) FM applied on backbone features with $\alpha_{b1}$.

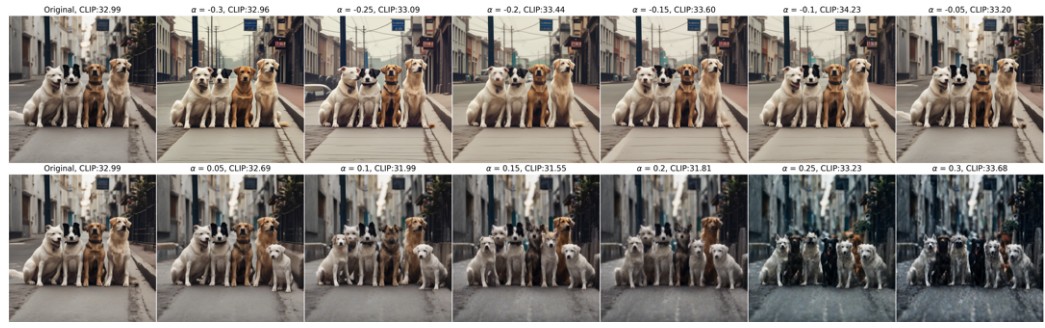

(b) FM applied on backbone features with $\alpha_{b2}$.

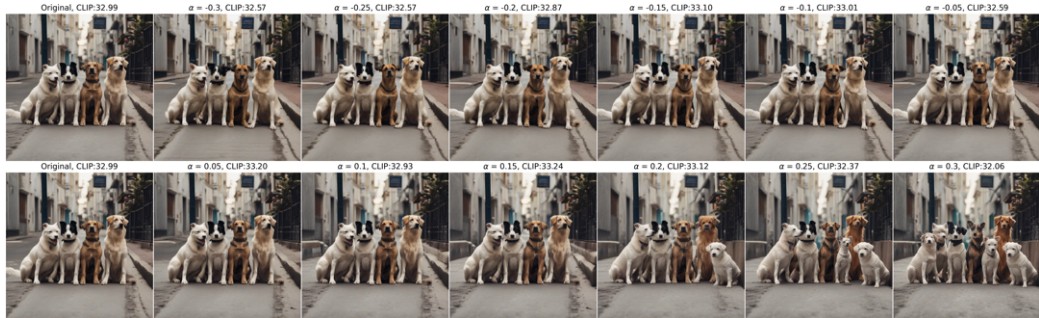

(c) FM applied on skip features in skip connections with $\alpha_{s1}$.

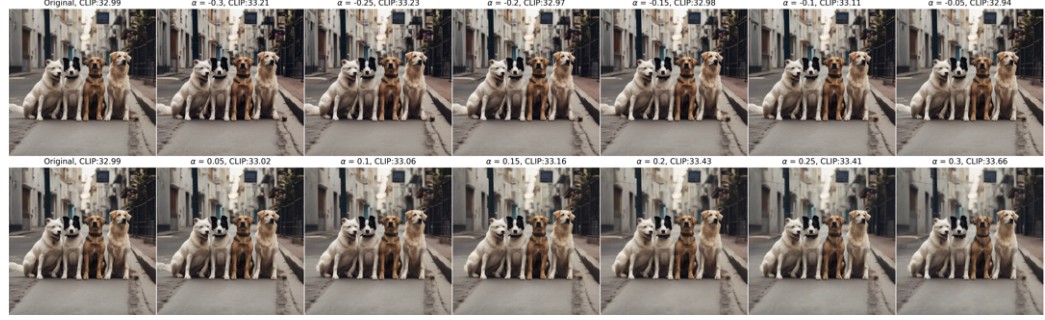

(d) FM applied on skip features in skip connections with $\alpha_{s2}$.

Figure I16: Prompt: *Five dogs on the street*.

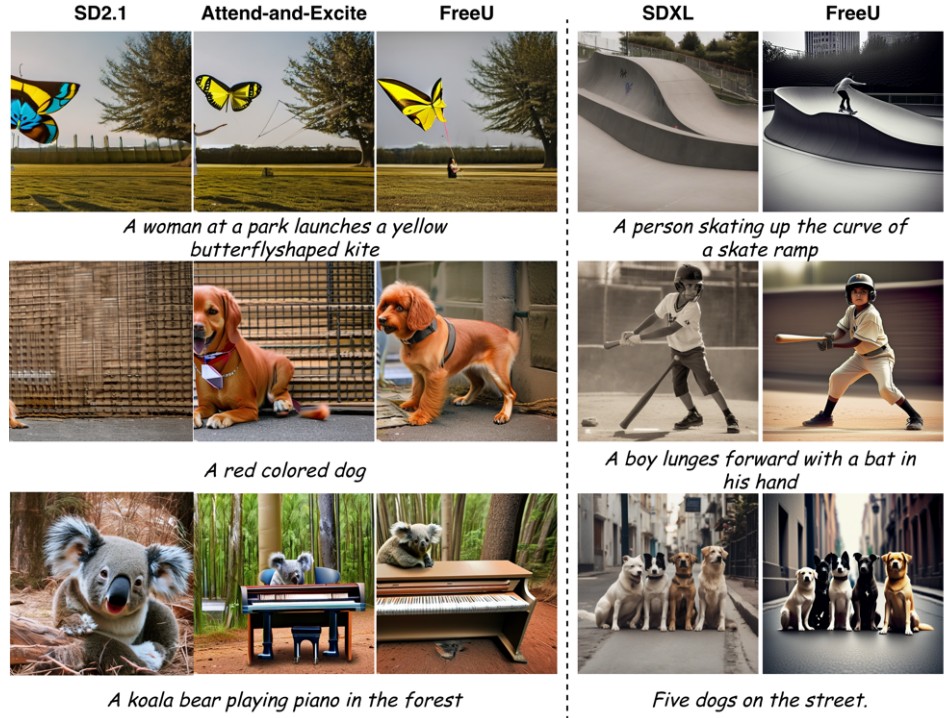

Figure J17: Visual comparison against baseline methods (FreeU and Attend-and-Excite) on SD2.1 and SDXL. The columns display the results generated using the prompts from Fig. 8 and Fig. 7.

## N    LLM INSTRUCTION FOR EVALUATION

Here we provide the full test prompt for evaluation on GPT-4o in Table N6. With these detailed LLM instructions, GPT-4o has shown strong capability in justifying the differences between the original images and those generated with FM.

## O    ANNOTATED DATASET FOR LLM TESTING

To evaluate the capability of GPT-4o using the instructions introduced in Appendix N, we manually curated an annotated dataset of 77 image pairs. Of these, 47 are *positive* samples (FM-processed images show improvement) and 30 are *negative* samples (no improvement); within the negative set, 20 pairs are ambiguous and 10 pairs exhibit degradation after FM. The 47 positive pairs include 24 examples shown in Figs. 1, 6, and 7, as well as 23 additional pairs selected from SDXL outputs. We assign label 0 to positive samples and label 1 to negative samples. Example JSON annotations are provided below in Tab. O7. The full dataset and annotations will be released upon acceptance.

## P    CUSTOMIZED PROMPT SET

In addition to the DRAWBENCH and COCO benchmarks, we evaluate on a small, customized list of 29 prompts, several of which are used for figures in this paper. Table P8 shows a selection of these prompts. The complete list will be provided in the Appendix and made publicly available upon acceptance.

## Q    ABLATION STUDY ON RATIO $\gamma$ FOR CHANNEL SELECTION.

In addition to the ablation study presented in the context, we further provide an ablation study on the channel selection ratio of $\gamma$. As shown in Fig. Q20, $\gamma$ controls the proportion of channels to which

**System Prompt:**

You are an expert image evaluator. You will be shown two separate images generated from the same text prompt—

an Original Image and a Modified Image.

Your task is to evaluate four specific aspects and determine if the Modified Image is better than the Original Image for each criterion.

For each criterion, first respond with "Yes" (if ANY part of the Modified Image is better than the Original Image) or "No" (for all other situations).

If you answer "No", then also answer a follow-up question:

"Is the modified image clearly worse or is it similar and you just don't know which one is better?"

Respond with "no" (if clearly worse) or "similar" (if you don't know).

Format your response exactly as follows:

INTEGRITY: [Yes/No] [no/similar if No]

NUMERICAL_ACCURACY: [Yes/No] [no/similar if No]

COMPLETENESS: [Yes/No] [no/similar if No]

VISUAL_REFINEMENT: [Yes/No] [no/similar if No]

**User Prompt:**

Prompt: {prompt}

Please evaluate the Modified Image compared to the Original Image on these four criteria:"

1. **INTEGRITY (No Crashes)**: Look for missing regions, heavy artifacts, or visible "tearing" where pixels appear incomplete or chopped off.

Does the Modified Image show ANY improvement in visual integrity compared to the Original Image?

If No, is the Modified Image clearly worse or is it similar?

2. **NUMERICAL_ACCURACY**: If the prompt contains explicit numeric values (e.g. "5 apples," "3 windows"),

does the Modified Image show ANY improvement in depicting the specified counts or measurements compared to the Original Image, even if partial?

If No, is the Modified Image clearly worse or is it similar?

3. **COMPLETENESS (Complex Prompts)**: For prompts listing multiple objects, actions, or fine details,

does the Modified Image show ANY improvement in including required elements (positions, relationships, attributes) compared to the Original Image?

If No, is the Modified Image clearly worse or is it similar?

4. **VISUAL_REFINEMENT (Color & Quality)**: Assess overall color consistency, richness, contrast, sharpness, and noise level.

Does the Modified Image show ANY improvement in visual quality compared to the Original Image?

If No, is the Modified Image clearly worse or is it similar?

Respond with the exact format specified above.

**In-context Examples**

Input: {USER PROMPT}

Output:

Table N6: Full LLM instruction for evaluation.

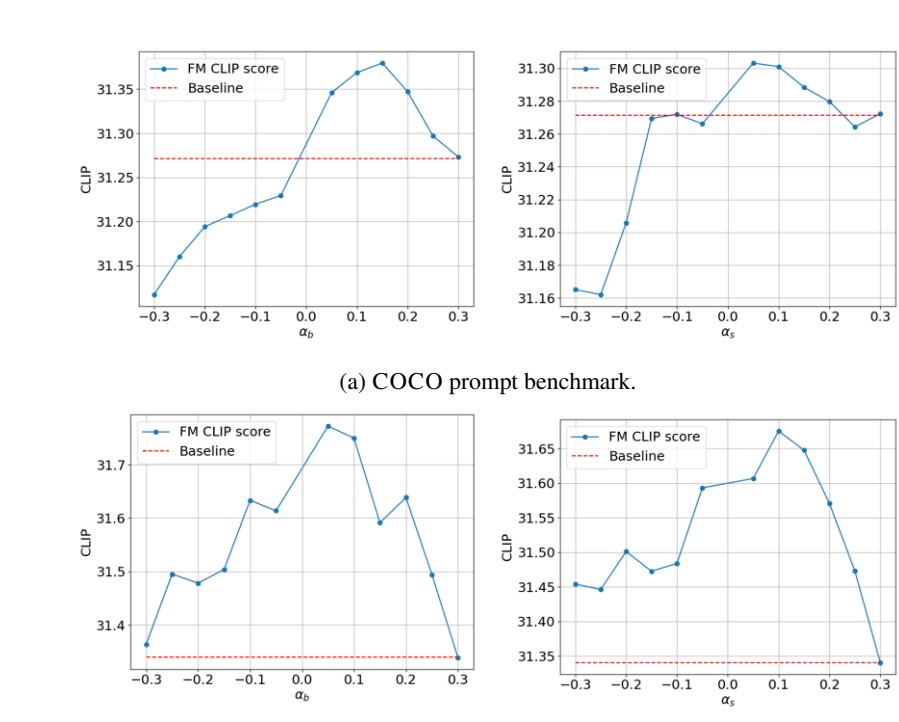

(a) COCO prompt benchmark.

(b) DRAWBENCH prompt benchmark.

Figure K18: CLIP score results on two standard benchmarks: COCO (top) and DRAWBENCH (bottom) using SDXL.

```
{
  "prompt": "Two dogs on the street",
  "images": {
    "0": "./FM_annotated_images/original_slide1_image0.png",
    "1": "./FM_annotated_images/original_slide1_image1.png"
  },
  "label": "1",
},
{
  "prompt": "Two dogs on the street",
  "images": {
    "0": "./FM_annotated_images/original_slide2_image2.png",
    "1": "./FM_annotated_images/original_slide2_image3.png"
  },
  "label": "1",
},
                                    ⋮
{
  "prompt": "Two cats and one dog sitting on the grass",
  "images": {
    "0": "./FM_annotated_images/original_slide3_image4.png",
    "1": "./FM_annotated_images/original_slide3_image5.png"
  },
  "label": "1",
}
```

Table O7: Annotations Examples.

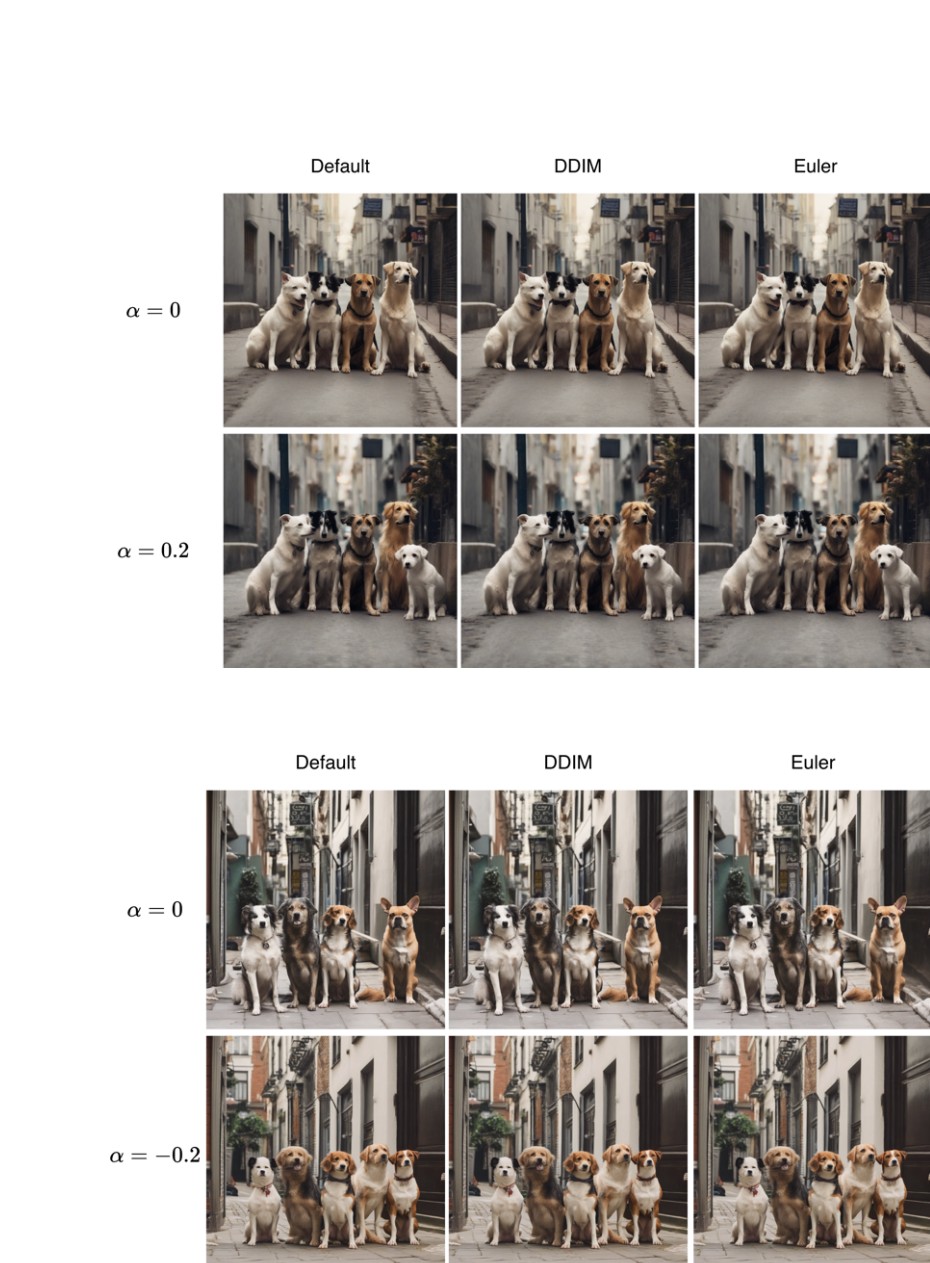

Figure M19: SDXL results under different seeds and samplers. Prompt: Five dogs on the street. Results on the top are generated using Seed 42, while results on the bottom are generated using Seed 300.

Two traffic signs on a post under a cloudy sky.

A long-haired Siamese cat sits in the living room, in front of the television.

A road sign and street light are on a light pole.

The tip of a horse's head can be seen and beyond that is a busy,

little lane with cars and shops.

A man in a blue apron at a juice vending cart.

A tie with a photograph of a race car repeated up and down as the design.

A living room with a pink and white striped couch next to a flat screen TV.

Street signs on a metal pole outside of a building.

$\vdots$

Table P8: Custom prompt examples (only a subset shown).

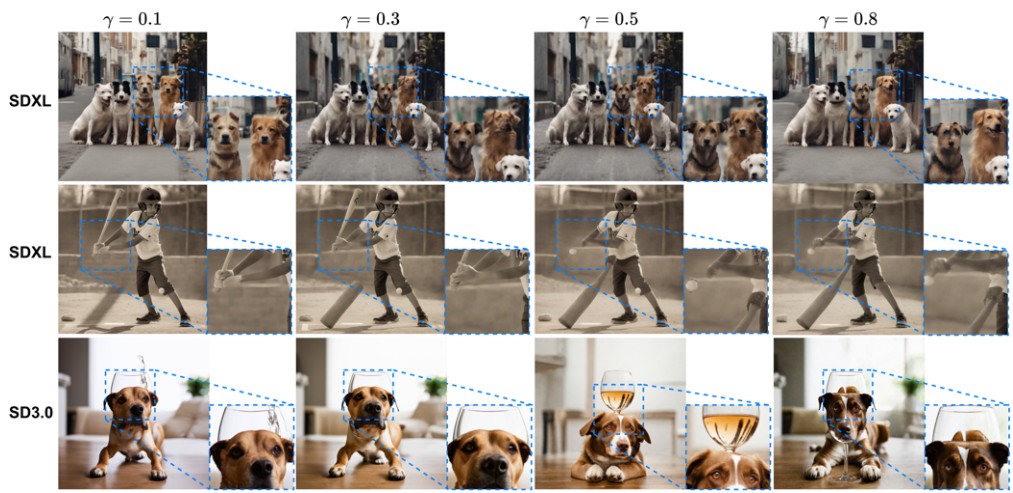

Figure Q20: Ablation study on the channel selection ratio $\gamma$ for FM. Zoomed-in differences highlight the impact of varying $\gamma$. First row and second row are the results of SDXL. The results of SD3.0 are in third row.

FM is applied. We evaluate the effect of different $\gamma$ between 0.1 and 0.8. For diffusion models based on U-Net, a small $\gamma$ leads to significant improvements. In contrast, DiT-based models tend to benefit from larger $\gamma$ values, as FM is applied closer to the output layer. The zoomed-in regions in Fig. Q20 further highlight the visual differences resulting from different $\gamma$ settings.

# R    ABLATION STUDY ON APPLYING FM ON DIFFERENT LAYERS

In this section, we present additional visualizations comparing images generated by SDXL with and without FM. A vanilla Diffusion U-Net architecture comprises a primary backbone network—consisting of an encoder and a decoder—as well as skip connections that transfer information between corresponding layers. We conduct an ablation study on the modulation parameter $\alpha$ by applying FM independently at four different latent spaces within the U-Net, specifically targeting the deepest two layers containing backbone features and skip connections. As shown in Fig. R21, we use two modulation parameters for the backbone features ($\alpha_{b1}$ and $\alpha_{b2}$) and two for the skip connections ($\alpha_{s1}$ and $\alpha_{s2}$). Additional results are presented in Fig. R22, R23, R24, R25. All visualizations in this section are produced using a channel selection ratio of $\gamma = 0.1$, and the corresponding CLIP scores are provided as a reference.

For SD3 with the DiT backbone, our observations indicate that FM can only be reliably applied on the last MM-DiT block. When FM is applied to earlier blocks, the generated images tend to "crash," degenerating into noise and completely losing semantic structure. Therefore, for SD3 we restrict FM

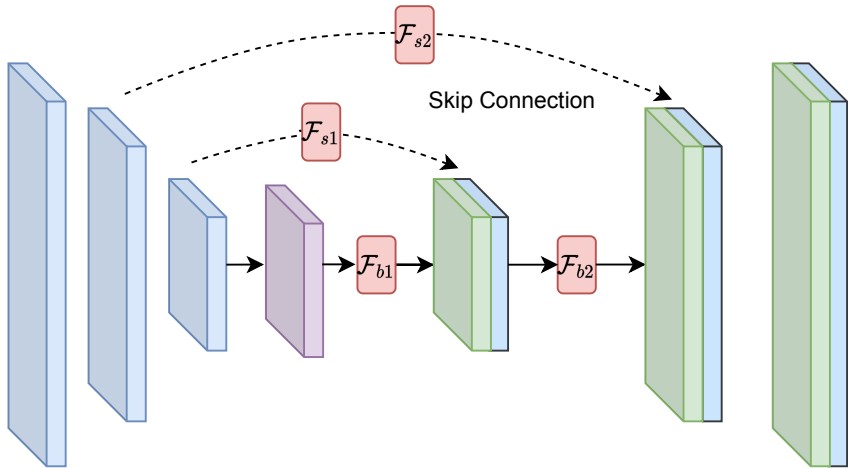

Figure R21: A simple visualization of modulation parameters in diffusion U-Net. For our U-Net architecture, we apply feature modulation using four parameters: $\alpha_{b1}$ in $\mathcal{F}_{b1}$ and $\alpha_{b2}$ in $\mathcal{F}_{b2}$ for backbone features at two different levels, and $\alpha_{s1}$ in $\mathcal{F}_{s1}$ and $\alpha_{s2}$ in $\mathcal{F}_{s2}$ for skip connection features.

to a single application at the modulation block of the last MM-DiT block, which preserves image fidelity while enabling controllable modulation.

## S  ABLATION STUDY ON CHANNEL SELECTION

In this section, we present additional ablation results on the channel selection strategy. Although the **Lemma** in Sec. 3.2 suggests that selecting the top 10% of channels with the highest mean values may provide better potential explainability, this choice is not strictly theoretically guaranteed. It remains unclear to what extent deeply fused features in the latent space align with semantic information. Therefore, we conduct further analysis and ablation experiments to examine alternative strategies.

Fig. S26 and Fig. S27 show visualization examples generated by SDXL under two different channel selection methods. We compare: (1) selecting channels based on the rank of their mean values (Fig. S26), which is the strategy adopted in the main results of this paper, and (2) selecting channels based on the rank of their variance (Fig. S27). For SDXL, which follows a U-Net architecture, we implemented up to four independent FM modules, each controlled by a distinct modulation parameter ($\alpha$) applied at different latent layers.

Our experiments show that even a small channel selection ratio $\gamma$ can substantially affect the final generated images. In particular, we found that setting $\gamma = 0.1$ in U-Net–based SD models achieves a balanced trade-off: it enables meaningful modulation while avoiding excessive alterations to the generated images. Considering these observations, we adopt the mean-based channel selection strategy as the default choice in this work.

## T  ADDITIONAL VISUALIZATIONS ON SD3.0

This section presents additional visual comparisons between original SD3.0 generated images and those enhanced by FM across different values of the modulation parameter $\alpha$. All images included in this section are generated using a channel ratio of $\gamma = 0.5$. The CLIP score is also presented as a reference. Results are shown in Fig. T28, T29, T30, T31, T32 and T33.

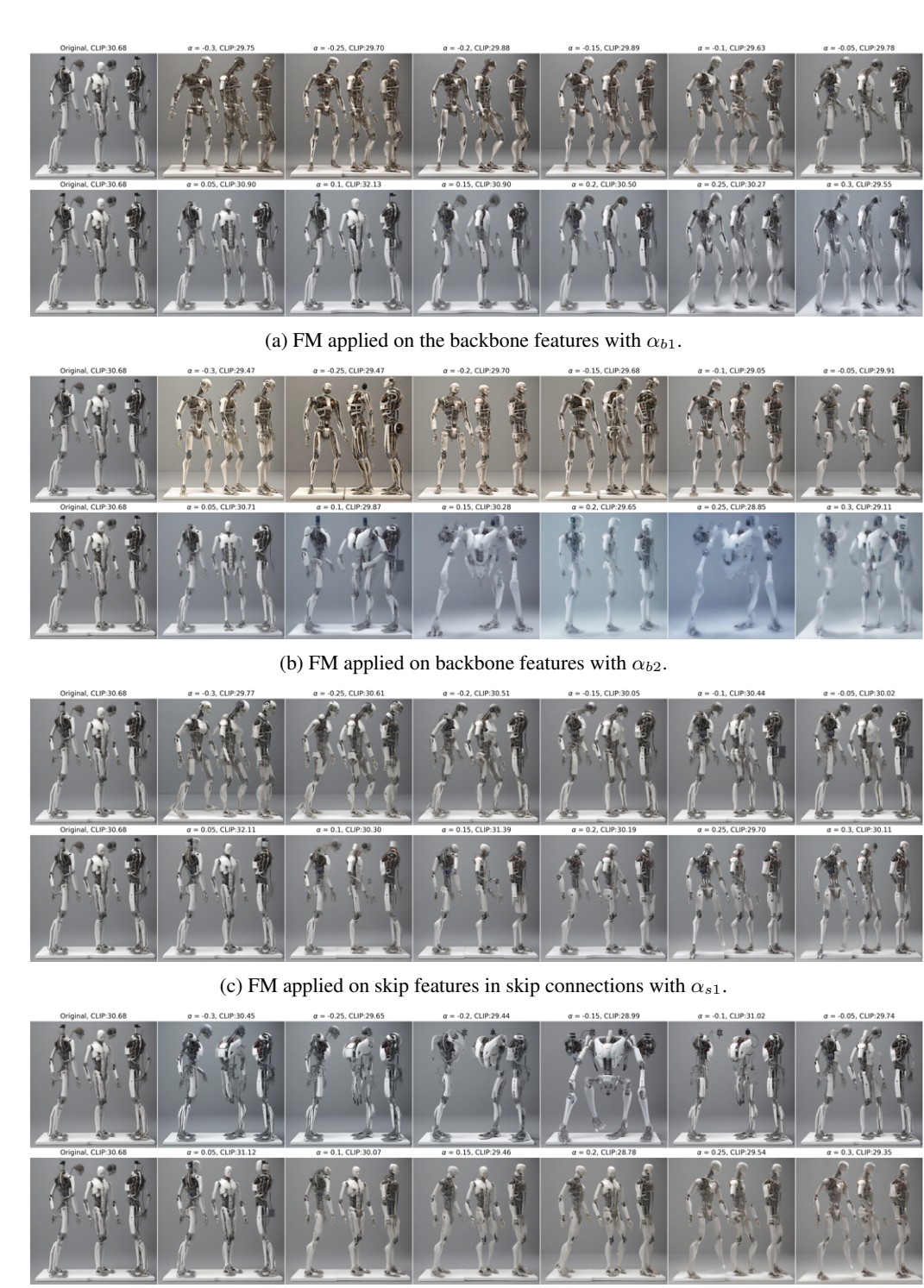

(a) FM applied on the backbone features with $\alpha_{b1}$.

(b) FM applied on backbone features with $\alpha_{b2}$.

(c) FM applied on skip features in skip connections with $\alpha_{s1}$.

(d) FM applied on skip features in skip connections with $\alpha_{s2}$.

Figure R22: Prompt: *A machine resembling a human being and able to replicate certain human movements and functions automatically.*

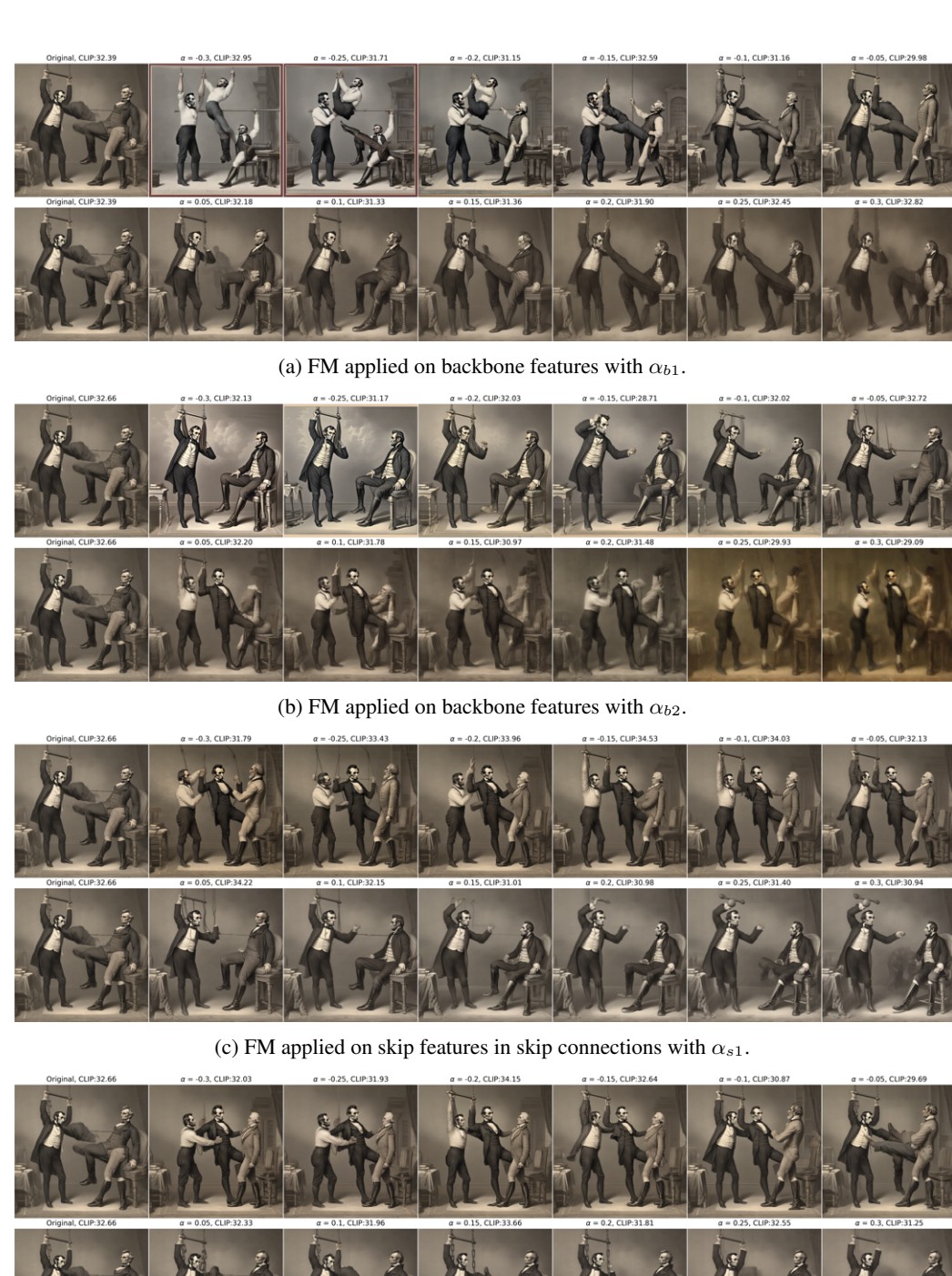

(a) FM applied on backbone features with $\alpha_{b1}$.

(b) FM applied on backbone features with $\alpha_{b2}$.

(c) FM applied on skip features in skip connections with $\alpha_{s1}$.

(d) FM applied on skip features in skip connections with $\alpha_{s2}$.

Figure R23: Prompt: *Abraham Lincoln touches his toes while George Washington does chin-ups. Lincoln is barefoot, Washington is wearing boots.*

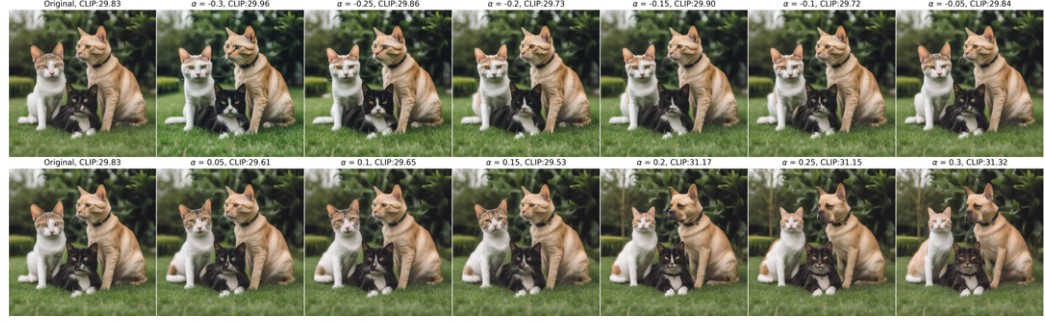

(a) FM applied on backbone features with $\alpha_{b1}$.

(b) FM applied on backbone features with $\alpha_{b2}$.

(c) FM applied on skip features in skip connections with $\alpha_{s1}$.

(d) FM applied on skip features in skip connections with $\alpha_{s2}$.

Figure R24: Prompt: *Two cats and one dog sitting on the grass*.

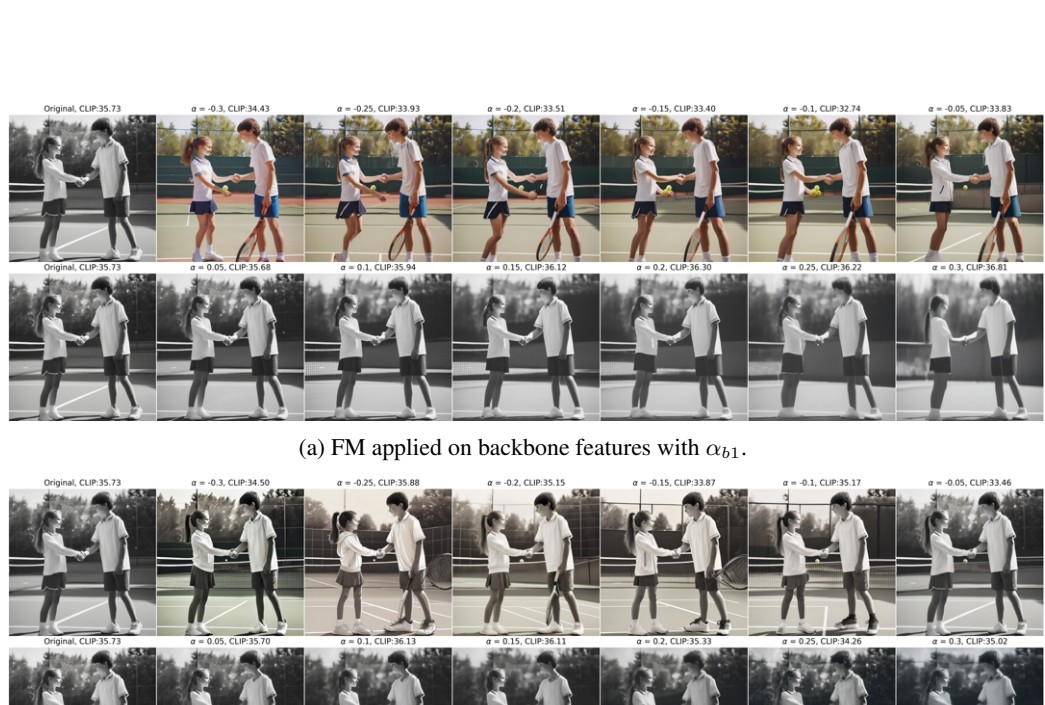

(a) FM applied on backbone features with $\alpha_{b1}$.

(b) FM applied on backbone features with $\alpha_{b2}$.

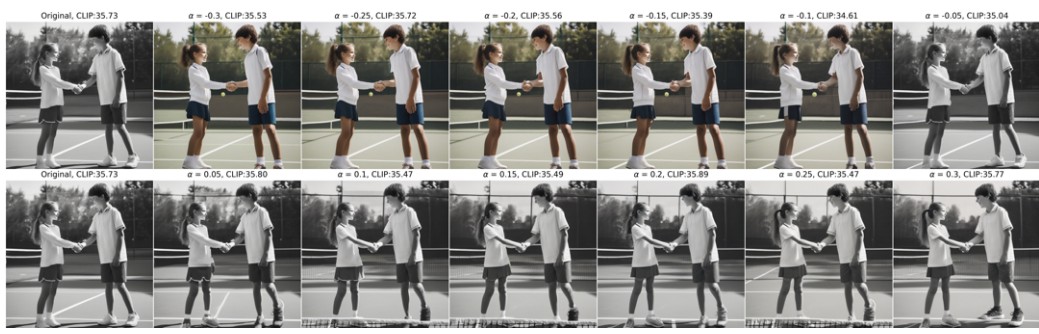

(c) FM applied on skip features in skip connections with $\alpha_{s1}$.

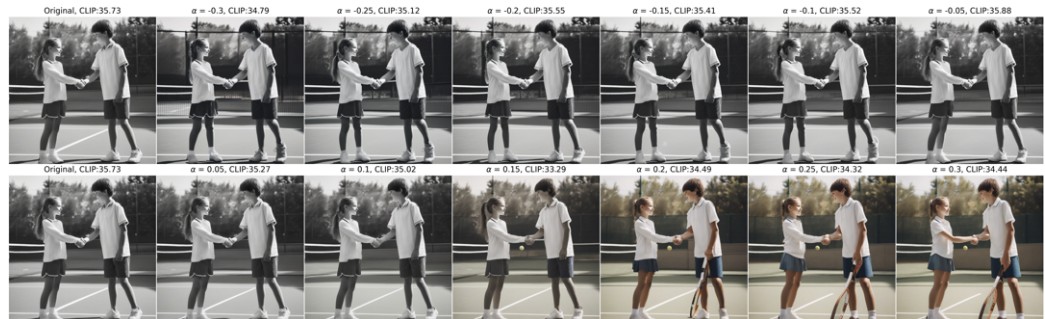

(d) FM applied on skip features in skip connections with $\alpha_{s2}$.

Figure R25: Prompt: *A girl and a boy is shaking hands on the tennis court*.

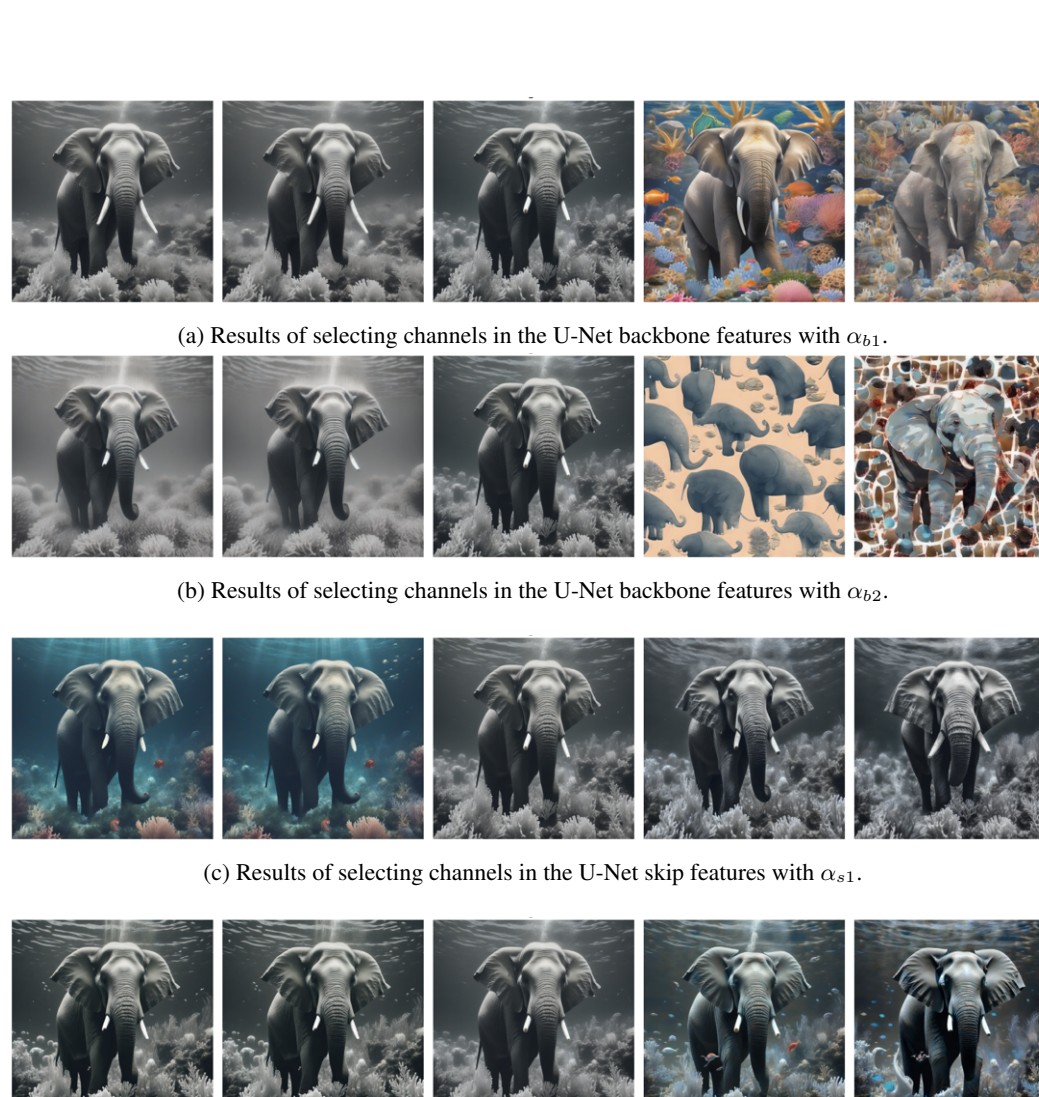

(a) Results of selecting channels in the U-Net backbone features with $\alpha_{b1}$.

(b) Results of selecting channels in the U-Net backbone features with $\alpha_{b2}$.

(c) Results of selecting channels in the U-Net skip features with $\alpha_{s1}$.

(d) Results of selecting channels in the U-Net skip features with $\alpha_{s2}$.

Figure S26: Prompt: *An elephant under the sea.* The original image is shown in the center. The four subfigures display the results of our ablation study on channel selection based on **Mean**. In the left two images, channels are selected in ascending order (from low to high mean values) using channel ratios of $\gamma = 0.05$ (second column) and $\gamma = 0.1$ (first column), respectively. In the right two images, channels are selected in descending order (from high to low) with $\gamma = 0.05$ (fourth column) and $\gamma = 0.1$ (fifth column), respectively. In all cases, the modulation parameter $\alpha$ is set to 0, meaning that FM is inactive and the selected channels are simply replaced by the mean values of the unselected channels.

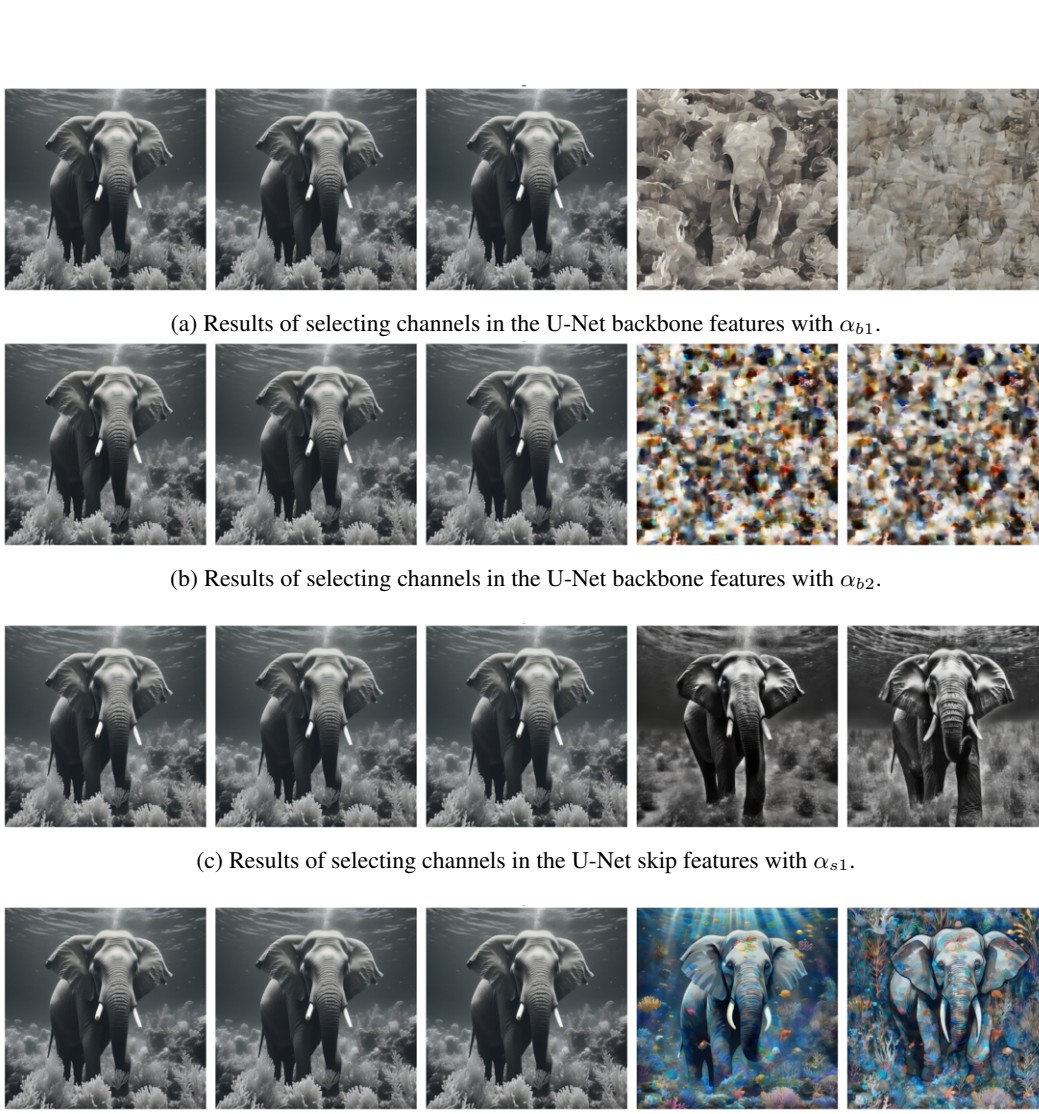

(a) Results of selecting channels in the U-Net backbone features with $\alpha_{b1}$.

(b) Results of selecting channels in the U-Net backbone features with $\alpha_{b2}$.

(c) Results of selecting channels in the U-Net skip features with $\alpha_{s1}$.

(d) Results of selecting channels in the U-Net skip features with $\alpha_{s2}$.

Figure S27: Prompt: *An elephant under the sea.* The original image is shown in the center. The four subfigures display the results of our ablation study on channel selection based on **Variance**. In the left two columns, channels are selected in ascending order (from low to high mean values) using channel ratios of $\gamma = 0.05$ (second column) and $\gamma = 0.1$ (first column), respectively. In the right two images, channels are selected in descending order (from high to low) with $\gamma = 0.05$ (fourth column) and $\gamma = 0.1$ (fifth column), respectively. In all cases, the modulation parameter $\alpha$ is set to 0, meaning that FM is inactive and the selected channels are simply replaced by the mean values of the unselected channels.

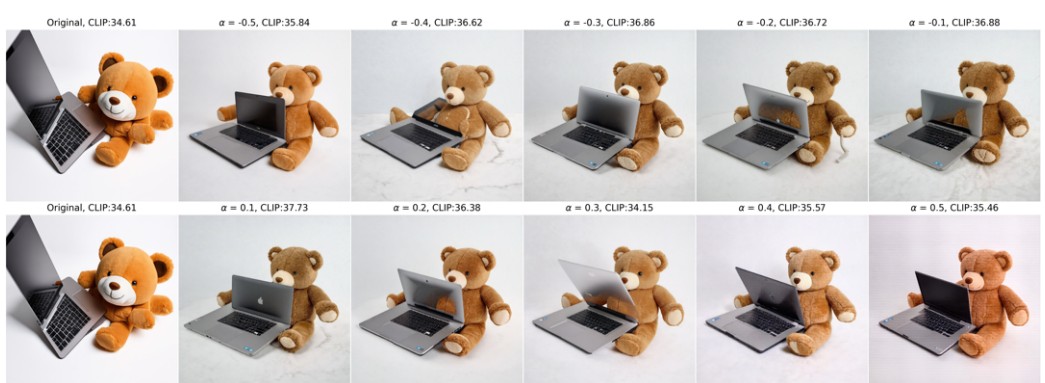

Figure T28: Prompt: *A laptop on top of a teddy bear*.

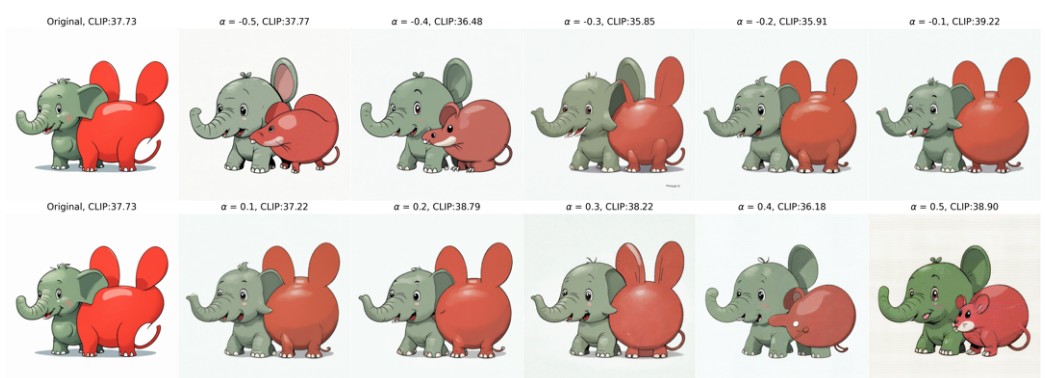

Figure T29: Prompt: *An illustration of a small green elephant standing behind a large red mouse*.

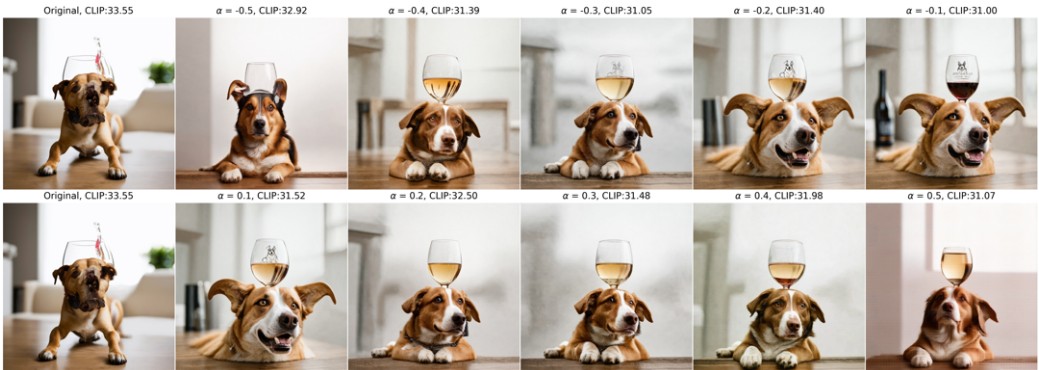

Figure T30: Prompt: *A wine glass on top of a dog*.

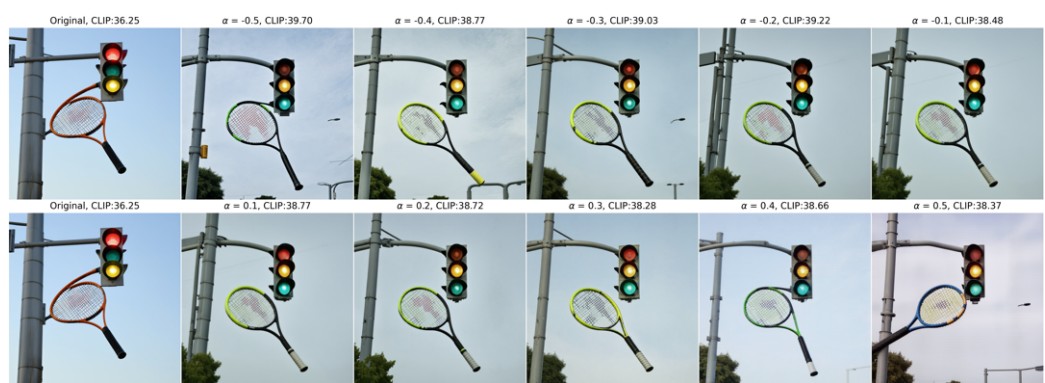

Figure T31: Prompt: *A tennis racket underneath a traffic light*.

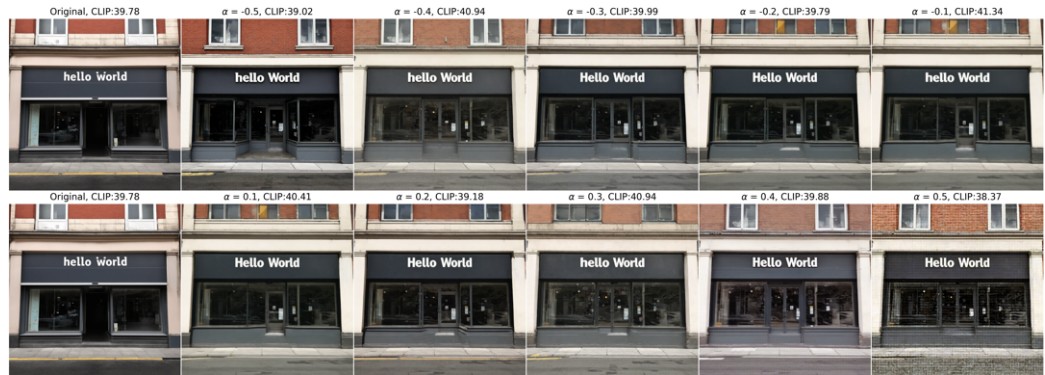

Figure T32: Prompt: *A storefront with Hello World written on it*.

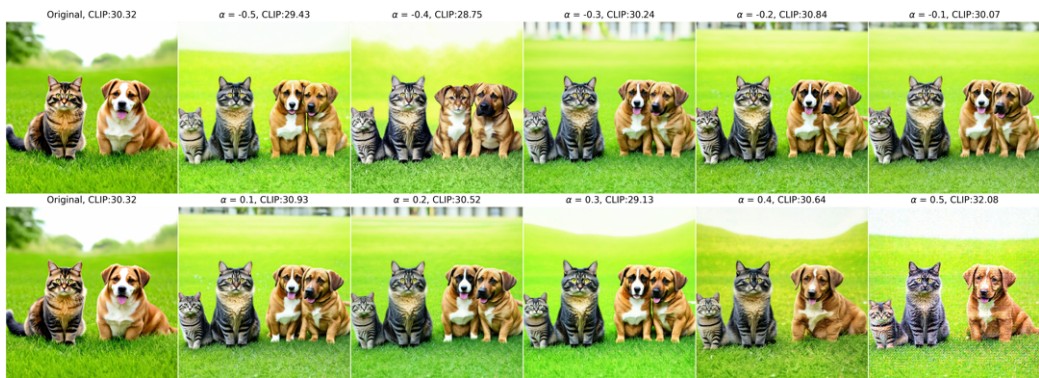

Figure T33: Prompt: *Two cats and two dogs sitting on the grass*.

