# OpenReview forum: "Feature Modulating for Diffusion Models"
_ICLR.cc/2026/Conference — Submitted to ICLR 2026_

### Official Review · Reviewer_rTb9 · 2025-10-24

**Soundness:** 2
**Presentation:** 2
**Contribution:** 2
**Rating:** 2
**Confidence:** 4

**Summary:**

This paper introduces a training-free method to enhance the image quality and text-image alignment of text-to-image diffusion models. The core contribution is a feature modulation technique that operates in a channel-wise manner on the internal features of the diffusion model, requiring no additional training or auxiliary inputs.

**Strengths:**

- The proposed method is training-free, which is a good advantage. This makes it computationally efficient, versatile, and easy to integrate into existing pre-trained diffusion models without incurring expensive training costs.
- The approach is straightforward and self-contained, operating directly on the model's features without the need for external training of modules or auxiliary inputs beyond the standard text prompt.

**Weaknesses:**

- The experiments are confined to Stable Diffusion. To demonstrate the generalizability and robustness of the proposed method, it is crucial to evaluate its performance across a broader range of modern T2I architectures, such as Flux, SANA, or Qwen-Image. This would strengthen the paper's claims.
- The method's performance appears highly sensitive to the hyperparameter $\alpha$, as shown in the ablation study. However, the paper provides no clear guidance on how to select an optimal value for $\alpha$. This raises concerns about the method's practical usability, as users might need to perform extensive trial-and-error to achieve good results. A more thorough analysis of $\alpha$'s behavior or a proposed strategy for its selection is needed.
- The paper omits comparisons with other relevant training-free methods that aim to improve generation quality, such as FreeU (which is referred in the related work section). A quantitative and qualitative comparison against such established baselines is essential to properly contextualize the proposed method's contributions and effectiveness.
- (minor) The figure reference seems to be wrong. In the sentence in line 82-84, the Figure seems to be Fig. 1 rather than Fig. 2.

**Questions:**

- Could the authors perform experiment regarding the applicability of this method to different model architectures?
- The results in Figure 8 show that the optimal value for $\alpha$ varies significantly across different samples (e.g., $\alpha=0.1$ is best for the first row, while negative values are better for the second). How should a user determine the best $\alpha$ for an arbitrary prompt? Have the authors considered any dynamic or sample-adaptive techniques for setting this value automatically?
- The evaluation relies heavily on an LLM-based assessment. While insightful, it would be beneficial to supplement this with other widely-used metrics for text-image alignment. Have the authors considered reporting metrics such as CLIP score or VQA score to provide a more comprehensive quantitative evaluation?
- Additionally, the paper's claim of enhancing image quality is not substantiated by quantitative analysis. Could the authors provide an evaluation using widely-used metrics, for example, CLIP image similarity or DINO scores, to measure the degree of improvement?

---

> ### Author Response · Authors · 2025-11-26
> **Response to Reviewer rTb9 (Part 1/3)**
>
> Dear Reviewer rTb9,
>
> Thank you for your constructive comments. We would like to answer your questions and concerns by following:
>
> >**(W1) The experiments are confined to Stable Diffusion. To demonstrate the generalizability and robustness of the proposed method, it is crucial to evaluate its performance across a broader range of modern T2I architectures, such as Flux, SANA, or Qwen-Image. This would strengthen the paper's claims.**
>
> (R1) We update the Appendix H ("FM on More Model Architectures"), which further includes three suggested architectures—Flux, SANA, and Qwen-Image. Please refer to A1 for a detail reply.
>
> >**(W2) The method's performance appears highly sensitive to the hyperparameter $\alpha$, as shown in the ablation study. However, the paper provides no clear guidance on how to select an optimal value for $\alpha$. This raises concerns about the method's practical usability, as users might need to perform extensive trial-and-error to achieve good results. A more thorough analysis of $\alpha$'s behavior or a proposed strategy for its selection is needed.**
>
> (R2) We show that only a small grid range of $\alpha$ is needed for searching. Please refer to A2 for a detail reply.
>
> >**(W3) The paper omits comparisons with other relevant training-free methods that aim to improve generation quality, such as FreeU (which is referred in the related work section). A quantitative and qualitative comparison against such established baselines is essential to properly contextualize the proposed method's contributions and effectiveness.**
>
> (R3) we add a T2I-CompBench evaluation in Sec.4.4, "Evaluation on T2I-CompBench", and Tab.4, which includes CLIP-based components but also measures multiple compositional aspects (color, shape, texture, spatial relations, numeracy). Please refer to A3 for a detail answer.
>
> >**(minor) The figure reference seems to be wrong. In the sentence in line 82-84, the Figure seems to be Fig. 1 rather than Fig. 2.**
>
> (R4) Thank you for pointing this out. We have corrected this in the revision.

---

> ### Author Response · Authors · 2025-11-26
> **Response to Reviewer rTb9 (Part 2/3)**
>
> >**(Q1) Could the authors perform experiment regarding the applicability of this method to different model architectures?**
>
> (A1) Yes. Our method is intentionally architecture-agnostic: it only requires access to intermediate feature maps and applies a simple per-channel reparameterization, without modifying training or model parameters. In this revision we additionally run FM on three more architectures—Flux, SANA, and Qwen-Image—using the same SD3 configuration and the same prompts as in Fig.H12 and Fig.H13 (with channel ratio $\gamma=0.5$ same as SD3), and report the visual results in the new Appendix H ("FM on More Model Architectures") with Fig.H14 and Fig.H15.
>
> >**(Q2) The results in Figure 8 show that the optimal value for $\alpha$ varies significantly across different samples (e.g., $\alpha=0.1$ is best for the first row, while negative values are better for the second). How should a user determine the best for an arbitrary prompt? Have the authors considered any dynamic or sample-adaptive techniques for setting this value automatically?**
>
> (A2) Most useful settings fall in the small grid $\{-0.2,-0.1,0.1,0.2\}$ within the safe range $[-0.3,0.3]$, so users only need to try a few values rather than search a wide continuum. This grid is not meant to identify a single *optimal* $\alpha$, but rather to provide a small range of values that reliably improve difficult cases and let the user pick a preferred setting from a compact option pool. Instead of exhaustively finding a global optimum, the practical goal is simply to find an $\alpha$ that improves the baseline image; this is reflected in our LLM evaluation in Tab.3, where the first improving value is almost always $\alpha=-0.2$ or $\alpha=-0.1$, so one or two extra FM samples typically suffice.
>
>   | Dataset   | $\alpha$ set                  |   -0.2           |   -0.1           |    0.1           |    0.2           |
>   |-----------|------------------------------|------------------|------------------|------------------|------------------|
>   | Custom    | $\{-0.2,\,0.2\}$             | 17 (85.00\%)     |   –              |   –              |  3 (15.00\%)     |
>   | Custom    | $\{-0.2,-0.1,0.1,0.2\}$      | 15 (62.50\%)     |  6 (25.00\%)     |  2 ( 8.33\%)     |  1 ( 4.17\%)     |
>   | DrawBench | $\{-0.2,\,0.2\}$             | 130 (84.97\%)    |   –              |   –              | 23 (15.03\%)     |
>   | DrawBench | $\{-0.2,-0.1,0.1,0.2\}$      | 116 (69.46\%)    | 26 (15.57\%)     | 15 ( 8.98\%)     | 10 ( 5.99\%)     |
>   | COCO      | $\{-0.2,\,0.2\}$             | 355 (66.85\%)    |   –              |   –              | 176 (33.15\%)    |
>   | COCO      | $\{-0.2,-0.1,0.1,0.2\}$      | 367 (52.28\%)    | 156 (22.22\%)    | 128 (18.23\%)    | 51 ( 7.26\%)     |
>
> FM is only needed when the baseline image is unsatisfactory, so $\alpha$ effectively acts as a simple user-controlled knob rather than a parameter that must always be tuned. Automatically predicting a globally optimal $\alpha$ for each prompt is an interesting direction for future work; in practice, however, once the user has found any $\alpha$ that clearly improves the image, the choice between several improving values is largely subjective (e.g., stronger object emphasis vs.\ more local detail), which is why we treat $\alpha$ as a user-facing control rather than a learned parameter in this work.

---

> ### Author Response · Authors · 2025-12-02
> **Response to Reviewer rTb9 (Part 3/3)**
>
> >**(Q3) The evaluation relies heavily on an LLM-based assessment. While insightful, it would be beneficial to supplement this with other widely-used metrics for text-image alignment. Have the authors considered reporting metrics such as CLIP score or VQA score to provide a more comprehensive quantitative evaluation?**
>
> (A3) We have experimented with CLIP scores and found that they can be misleading in our setting: if we allow per-image search over $\alpha$, it is easy to drive CLIP scores very high even when human raters (or GPT-4o) do not perceive the resulting images as better Instead, in this revision we add a T2I-CompBench evaluation in Sec.4.4, "Evaluation on T2I-CompBench", and Tab.4, which includes CLIP-based components but also measures multiple compositional aspects (color, shape, texture, spatial relations, numeracy). Using the official benchmark code and default settings for SD2.1 and SDXL, we report side-by-side results for the baselines, FreeU, Attend-and-Excite, and FM, showing that FM consistently improves over the baselines and is competitive with or better than these training-free methods across most metrics. We believe this benchmark provides a more informative and robust quantitative complement to the GPT-4o-based assessment than raw CLIP scores alone.
>
> | Method            | Backbone | Color         | Shape         | Texture       | Numeracy       | 3D            | Non-spat.      | Spat.         |
> |-------------------|----------|---------------|---------------|---------------|----------------|---------------|----------------|---------------|
> | Baseline          | SD2.1    | 0.40±0.35     | 0.39±0.30     | 0.47±0.32     | 0.34±0.23      | 0.23±0.16     | 0.30±0.03      | 0.05±0.21     |
> |                   | SDXL     | 0.60±0.36     | 0.48±0.31     | 0.54±0.37     | 0.49±0.24      | 0.37±0.18     | 0.31±0.03      | 0.23±0.40     |
> | FreeU             | SD2.1    | 0.58±0.14     | 0.57±0.14     | 0.55±0.15     | 0.46±0.27      | 0.34±0.17     | 0.31±0.03      | 0.19±0.37     |
> |                   | SDXL     | 0.57±0.15     | 0.52±0.15     | 0.51±0.14     | 0.44±0.25      | 0.31±0.03     | 0.31±0.03      | 0.22±0.39     |
> | Attend-and-Excite | SD2.1    | 0.49±0.15     | 0.48±0.15     | 0.47±0.15     | 0.45±0.25      | 0.33±0.18     | 0.31±0.03      | 0.18±0.36     |
> | FM                | SD2.1    | 0.58±0.33     | 0.58±0.30     | 0.58±0.32     | 0.49±0.23      | 0.37±0.17     | 0.31±0.03      | 0.21±0.30     |
> |                   | SDXL     | **0.73±0.30** | **0.62±0.30** | **0.68±0.34** | **0.63±0.25**  | **0.48±0.20** | **0.32±0.03**  | **0.31±0.44** |
>
> >**(Q4) Additionally, the paper's claim of enhancing image quality is not substantiated by quantitative analysis. Could the authors provide an evaluation using widely-used metrics, for example, CLIP image similarity or DINO scores, to measure the degree of improvement?**
>
> (A4) We add a T2I-CompBench evaluation in Sec.4.4, "Evaluation on T2I-CompBench", and Tab.4. T2I-CompBench is a composite benchmark that aggregates several widely used metrics, including CLIP-based image–text similarity, BLIP-VQA scores, and detection-based consistency. To make this more explicit, we report below the SDXL scores for the three components that most directly reflect image quality and text–image consistency, comparing the baseline SDXL model and SDXL+FM:
>
> | Metric      | SDXL  | FM    | Improvement        |
> |------------|-------|-------|--------------------|
> | CLIP score | 0.3026| 0.3137| +0.0111 (+3.67%)   |
> | BLIP-VQA   | 0.7914| 0.8820| +0.0906 (+11.45%)  |
> | Detection  | 0.3642| 0.4744| +0.1102 (+30.27%)  |
>
> These three numbers are directly extracted from the T2I-CompBench decomposition: CLIP score measures global image–text similarity, BLIP-VQA approximates human question–answering about the generated image, and the detection component evaluates whether objects described in the prompt are correctly detected and localized.

---

### Official Review · Reviewer_sVfQ · 2025-10-29

**Soundness:** 1
**Presentation:** 3
**Contribution:** 1
**Rating:** 2
**Confidence:** 4

**Summary:**

The paper proposes a training‑free, channel‑wise feature modulation (FM) module that is inserted into the denoising steps of text‑to‑image diffusion models. After per‑channel min–max normalization, a monotone quadratic mapping $f(z) = z + \alpha z(1-z)$ is applied and then rescaled back; only the top‑$\gamma$ channels by mean activation are modulated. The method is intentionally simple (single scalar $\alpha$ and selection ratio $\gamma$), architecture‑agnostic, and is demonstrated on SD2.1 (U‑Net), SDXL (U‑Net), and SD3.0 (DiT).

**Strengths:**

+ Simple, training‑free mechanism that can be dropped into existing denoisers; no model updates or auxiliary inputs are required. The block diagram and per‑step placement are clear and easy to read.

+ Works on various models such as SD2.1, SDXL and SD3.0; the visuals in Figs. 6–7 (pp. 6–7) illustrate consistent qualitative gains on integrity, counts, and missing parts.

**Weaknesses:**

+ The core quantitative claim relies on a GPT‑4o prompt protocol (Table 1, 2 on p. 8). While the prompt is carefully designed and validated on a small annotated set, this remains non‑standard and potentially brittle to prompt choices. The CLIP‑score plots in Appendix C show only modest improvements and are not tied to the large‑scale improvement rates. A rigorous evaluation should include human A/B preference studies and automatic, task‑specific measures (e.g., count accuracy via detectors or segmentation; text‑image alignment with multiple metrics; robustness over seeds). This is the largest reason my score is conservative.

+ The method is training‑free, but the paper grid‑searches some $\alpha$ per prompt (Sec. 4.3, Table 2) and the Reproducibility Statement admits per‑case hyperparameter tuning is often needed. Yet the Conclusion claims “without incurring any additional computational costs.” These statements conflict so please quantify the overhead (extra function evaluations / wall‑clock) on to get the results in your benchmarks.

+ The paper cites FreeU, RB‑Modulation, Attend‑and‑Excite, etc., but does not systematically compare against them on the same prompts/metrics. Without such baselines the significance of FM is hard to judge.

+ For SD3.0, FM “crashes” earlier blocks and only works reliably at the last MM‑DiT block (Appendix J). This narrows the claim of being architecture‑agnostic; please analyze why and whether lighter‑weight normalizations or mixed‑precision clamps mitigate this.

**Questions:**

+ How many extra samples per prompt are needed on average to find a good $\alpha$? Please report compute overhead and win‑rate vs. cost trade‑off. Could you provide one‑shot heuristics (e.g., function of layer norms, prompt length, CFG scale) that pick reasonably good $\alpha$ without search?

+ Can you add side‑by‑side comparisons with FreeU, RB‑Modulation, Attend‑and‑Excite (or similar training‑free methods) on the same prompts using both the LLM judge and conventional metrics?

+ How stable are improvements across random seeds and samplers (DDIM/DPMSolver/etc.)?

+ Why does early‑layer modulation “crash” SD3.0? Any explanation that adjusting $gamma$ or adding per‑feature clipping avoids divergence?

---

> ### Author Response · Authors · 2025-11-26
> **Response to Reviewer sVfQ (Part 1/4)**
>
> Dear Reviewer sVfQ,
>
> Thank you for your insightful comments. We reply to your concerns and questions as follow:
>
> >**(W1) The core quantitative claim relies on a GPT‑4o prompt protocol (Table 1, 2 on p. 8). While the prompt is carefully designed and validated on a small annotated set, this remains non‑standard and potentially brittle to prompt choices. The CLIP‑score plots in Appendix C show only modest improvements and are not tied to the large‑scale improvement rates. A rigorous evaluation should include human A/B preference studies and automatic, task‑specific measures (e.g., count accuracy via detectors or segmentation; text‑image alignment with multiple metrics; robustness over seeds). This is the largest reason my score is conservative.**
>
> (R1) We add a comprehensive evaluation on T2I-Compbench benchmark with the comparison with other SOTA methods in Sec 4.4. Concretely, we report BLIP-VQA correctness scores for attribute binding (color, shape, texture), UniDet-based detection scores for object presence and spatial relations, and CLIP-based text--image similarity scores, which are further aggregated into the “Numeracy”, “3D”, “Non-spatial”, and “Spatial” composite metrics shown in Tab.4 in A2.
>
> >**(W2) The method is training‑free, but the paper grid‑searches some $\alpha$ per prompt (Sec. 4.3, Table 2) and the Reproducibility Statement admits per‑case hyperparameter tuning is often needed. Yet the Conclusion claims “without incurring any additional computational costs.” These statements conflict so please quantify the overhead (extra function evaluations / wall‑clock) on to get the results in your benchmarks.**
>
> (R2) We clarify our claim in the conclusion by "without introducing any additional training cost". We report the compute overhead in A1. Please refer to A1 for a detail answer.
>
> >**(W3) Section 2, related work, does not explain how it is related to the proposed work or how it differs from the proposed method.**
>
> (R3) We add a comprehensive evaluation on T2I-Compbench benchmark with the comparison with other SOTA methods in Sec 4.4. Please refer to A2 for detail answer and Tab.4.
>
> >**(W4) For SD3.0, FM “crashes” earlier blocks and only works reliably at the last MM‑DiT block (Appendix J). This narrows the claim of being architecture‑agnostic; please analyze why and whether lighter‑weight normalizations or mixed‑precision clamps mitigate this.**
>
> (R4) FM is architecture‑agnostic to different frameworks of diffusion models as it works for both U-Net and DiT structures. We further verified its effectiveness on a broader range of modern T2I architectures, including Flux, SANA, and Qwen-Image (Please refer to A1 for Reviewer rTb9). Please refer to A4 for the detail analysis of the reason causing the “crashes”

---

> ### Author Response · Authors · 2025-11-26
> **Response to Reviewer sVfQ (Part 2/4)**
>
> >**(Q1) How many extra samples per prompt are needed on average to find a good $\alpha$? Please report compute overhead and win‑rate vs. cost trade‑off. Could you provide one‑shot heuristics (e.g., function of layer norms, prompt length, CFG scale) that pick reasonably good $\alpha$ without search?**
>
> (A1) For a single FM modulation with a fixed $\alpha$，there is no extra cost. We measured the per-step computational cost on a single A100 GPU as follows:
>
>   | Model   | $\alpha$ | GFLOPs  | TFLOPs |
>   |---------|----------|---------|--------|
>   | SDXL    | 0        | 2991.02 | 2.9910 |
>   | SDXL+FM | -0.2     | 2991.02 | 2.9910 |
>
> We report the win‑rate vs. cost trade‑off for LLM evaluation in Tab.3. The table below summarizes, for each dataset in Tab.2 and for both search settings (two $\alpha$ values vs. four), how often each $\alpha$ is the *first* value that yields an improvement under GPT‑4o:
>
>   | Dataset   | $\alpha$ set                  |   -0.2           |   -0.1           |    0.1           |    0.2           |
>   |-----------|------------------------------|------------------|------------------|------------------|------------------|
>   | Custom    | $\{-0.2,\,0.2\}$             | 17 (85.00\%)     |   –              |   –              |  3 (15.00\%)     |
>   | Custom    | $\{-0.2,-0.1,0.1,0.2\}$      | 15 (62.50\%)     |  6 (25.00\%)     |  2 ( 8.33\%)     |  1 ( 4.17\%)     |
>   | DrawBench | $\{-0.2,\,0.2\}$             | 130 (84.97\%)    |   –              |   –              | 23 (15.03\%)     |
>   | DrawBench | $\{-0.2,-0.1,0.1,0.2\}$      | 116 (69.46\%)    | 26 (15.57\%)     | 15 ( 8.98\%)     | 10 ( 5.99\%)     |
>   | COCO      | $\{-0.2,\,0.2\}$             | 355 (66.85\%)    |   –              |   –              | 176 (33.15\%)    |
>   | COCO      | $\{-0.2,-0.1,0.1,0.2\}$      | 367 (52.28\%)    | 156 (22.22\%)    | 128 (18.23\%)    | 51 ( 7.26\%)     |
>
> In our GPT‑4o evaluation we therefore mimic a user who only turns the FM “knob” when the baseline is unsatisfactory. Concretely, we treat $\alpha$ as a 1D control knob and test it in a fixed small grid $\{-0.2,-0.1,0.1,0.2\}$ in a fixed order from $-0.2$ toward $0.2$, stopping as soon as GPT‑4o judges an FM image as improved. This means each prompt uses at most four extra FM samples on top of the baseline, and in practice most improved prompts require only one or two trials. Across all datasets and settings, the majority of improved prompts are already fixed by $\alpha=-0.2$, with most of the remaining wins coming from $\alpha=-0.1$.
>
> Compared to designing a one‑shot heuristic that predicts the best $\alpha$, our focus in this work is on the existence of an improving $\alpha$ in a very small discrete range: there can be multiple $\alpha$ values that produce better images than the baseline, so it is sufficient to search over a tiny grid. Our experiments show that searching only within $[-0.2,0.2]$ using the four discrete values ${-0.2,-0.1,0.1,0.2}$ is already enough to find such an improving $\alpha$ in most cases (see the revised Sec. 4.4 and Tab. 5). Developing heuristics that directly predict the optimal $\alpha$ is an interesting direction for future work.

---

> ### Author Response · Authors · 2025-11-26
> **Response to Reviewer sVfQ (Part 3/4)**
>
> >**(Q2) Can you add side‑by‑side comparisons with FreeU, RB‑Modulation, Attend‑and‑Excite (or similar training‑free methods) on the same prompts using both the LLM judge and conventional metrics?**
>
> (A2) We have added Appendix J and Fig.J17 for a visual compariason with the same prompts used in Fig.7 and Fig.8.
> Our LLM-based evaluation was carefully tailored to FM, so comparing other methods under the exact same LLM prompts would not be the fairest evaluation. Instead, we add a conventional side-by-side comparison on the standard T2I-CompBench, using the official benchmark code and default settings on SD2.1 and SDXL (Sec.4.4, Tab.4).
>
> | Method            | Backbone | Color         | Shape         | Texture       | Numeracy       | 3D            | Non-spat.      | Spat.         |
> |-------------------|----------|---------------|---------------|---------------|----------------|---------------|----------------|---------------|
> | Baseline          | SD2.1    | 0.40±0.35     | 0.39±0.30     | 0.47±0.32     | 0.34±0.23      | 0.23±0.16     | 0.30±0.03      | 0.05±0.21     |
> |                   | SDXL     | 0.60±0.36     | 0.48±0.31     | 0.54±0.37     | 0.49±0.24      | 0.37±0.18     | 0.31±0.03      | 0.23±0.40     |
> | FreeU             | SD2.1    | 0.58±0.14     | 0.57±0.14     | 0.55±0.15     | 0.46±0.27      | 0.34±0.17     | 0.31±0.03      | 0.19±0.37     |
> |                   | SDXL     | 0.57±0.15     | 0.52±0.15     | 0.51±0.14     | 0.44±0.25      | 0.31±0.03     | 0.31±0.03      | 0.22±0.39     |
> | Attend-and-Excite | SD2.1    | 0.49±0.15     | 0.48±0.15     | 0.47±0.15     | 0.45±0.25      | 0.33±0.18     | 0.31±0.03      | 0.18±0.36     |
> | FM                | SD2.1    | 0.58±0.33     | 0.58±0.30     | 0.58±0.32     | 0.49±0.23      | 0.37±0.17     | 0.31±0.03      | 0.21±0.30     |
> |                   | SDXL     | **0.73±0.30** | **0.62±0.30** | **0.68±0.34** | **0.63±0.25**  | **0.48±0.20** | **0.32±0.03**  | **0.31±0.44** |
>
> This provides a conventional, non-LLM metric complementing our GPT-4o-based evaluation and shows that FM consistently improves over the baselines and is competitive with baselines.
>
> >**(Q3) How stable are improvements across random seeds and samplers (DDIM/DPMSolver/etc.)?**
>
> (A3) We add results under different random seeds and samplers (including DDIM and Euler) in Appendix M (Fig. M19). From these examples, FM appears to depend mainly on the underlying image content rather than on the exact configuration. Changing the sampler only has a small effect: across samplers, FM produces similar improvements. In contrast, changing the random seed changes the entire generated image, so some additional tuning of $\alpha$ may be needed. However, we consistently find effective $\alpha$ values within our recommended discrete range $[-0.2,0.2]$. In the example of using seed 300, $\alpha=-0.2$ still improves the result in a similar way by correcting the number of dogs.

---

> ### Author Response · Authors · 2025-12-02
> **Response to Reviewer sVfQ (Part 4/4)**
>
> >**(Q4) Why does early‑layer modulation “crash” SD3.0? Any explanation that adjusting $\gamma$ or adding per‑feature clipping avoids divergence?**
>
> (A4) Empirically, on SD3.0 we observe that whenever FM is applied to any MM-DiT block other than the last one, the denoising process collapses: the final outputs become almost pure noise with no recognizable structure, even for small $|\alpha|$ in our usual range. We provide one possible explanation as follows.
>
> In DiT-based SD3.0, the denoising network at each step can be written as a deep composition of $L$ MM-DiT blocks,
> where $x_t \in \mathbb{R}^d$ and $x_{t-1}$ denote the latents before and after one denoising update, each $F_\ell$ denotes the transformation implemented by the $\ell$-th MM-DiT block, and $F = F_L \circ F_{L-1} \circ \cdots \circ F_1$ denotes the full denoising map at that step. One denoising step can thus be expressed schematically as
> $$x_{t-1} = F(x_t) = F_L \circ F_{L-1} \circ \cdots \circ F_1(x_t).$$
>
> Suppose FM is inserted into an early block indexed by $k$ (e.g., $k=1$), so that the intermediate feature at this location is $z_k = F_k \circ \cdots \circ F_1(x_t)$. Let $\Delta z$ denote the perturbation introduced by FM at $z_k$ (a function of $\alpha$ and the channel-selection ratio $\gamma$), and let $J_{F_\ell}$ be the Jacobian matrix of $F_\ell$ with respect to its input.
>
> Linearizing around the unmodulated trajectory gives the approximate change in the output latent
> $$\Delta x_{t-1} \approx J_{F_L}\,J_{F_{L-1}}\cdots J_{F_{k+1}}\,\Delta z.$$
> Because SD3.0 uses a long stack of MM-DiT blocks, the product of Jacobians
> $$J_{F_L}\,J_{F_{L-1}}\cdots J_{F_{k+1}}$$
> can have large singular values. In that case, even a modest per-channel perturbation $\Delta z$—coming from a small $|\alpha|$—may be strongly amplified as it propagates to the output, pushing $x_{t-1}$ far away from the denoising manifold. Once the trajectory leaves this manifold early in the chain, subsequent steps cannot recover, and the process converges to noisy, “crashed” samples. When FM is applied only in the last MM-DiT block, the effective Jacobian product reduces to $J_{F_L}$, so the amplification path is much shorter and we empirically obtain stable behavior.
>
> Regarding $\gamma$ and clipping, we have experimented with reducing the channel-selection ratio $\gamma$ on SD3.0, including very small values, but early-layer FM still led to the same kind of collapse, which is consistent with the above picture: shrinking $\gamma$ mainly rescales $\Delta z$ and does not remove the multiplicative amplification by the Jacobian product. We have not yet tried per-feature clipping as suggested by the reviewer. We appreciate this suggestion and will consider clipping-based stabilization strategies (e.g., bounding $\Delta z$ before it is propagated through the remaining blocks) as a promising direction for future work.

---

### Official Review · Reviewer_usGn · 2025-10-30

**Soundness:** 3
**Presentation:** 2
**Contribution:** 3
**Rating:** 4
**Confidence:** 4

**Summary:**

This paper proposed a simple channel-wise feature modulation approach to improve the generated image quality in latent diffusion models. The experiments are carried out on stable diffusion models and show the effectiveness by using a heuristic parameter with mean-based channel selection. Results are also validated with GPT-4o as a judge.

**Strengths:**

- The proposed modulation method is training-free and finding the parameter $\alpha$ is heuristic. Therefore, the proposed method is efficient.
- From the results in the paper, the generated images are significantly improved after the proposed modulation.
- The paper leverages GPT-4o for feasible evaluation.

**Weaknesses:**

- Finding the good enough $\alpha$ takes multiple tries and the most optimal $\alpha$ may not be guaranteed.
- The paper does not make a comparison with other modulation methods.
For example, the following works also propose feature modulation methods, although the focus is towards high-resolution images.

[1] https://arxiv.org/pdf/2411.18552

[2] https://www.nature.com/articles/s41598-025-94381-8

[3] https://openreview.net/forum?id=OIqOpdyhTd

- Section 2, related work, does not explain how it is related to the proposed work or how it differs from the proposed method.

Minor:
- The paper does not follow the ICLR template for tables. Table captions are supposed to be at the top.

- Writing should be improved. For example, there is a grammatical mistake in line 091, "are struggle with" should be "struggle with".

- Fig. 1 is not referenced in the main paper, but it is from the supplementary material.

- Table 1 spacing should be improved.

- Line 446, generally, it is better to use Figure at the beginning of the sentence rather than Fig.

- Section 4.1, implementation details are more like an experimental setup. It does not explain how the proposed method is implemented.

**Questions:**

- What is the resolution of the images in the experiments?
- What is the performance of FM if the resolution varies?
- What is the performance comparison with other modulation methods in terms of the criteria used in the paper?

---

> ### Author Response · Authors · 2025-11-26
> **Response to Reviewer usGn (Part 1/2)**
>
> Dear Reviewer usGn,
>
> Thank you for your insightful comments. We reply to your questions as follow:
>
> >**(W1) Finding the good enough $\alpha$ takes multiple tries and the most optimal $\alpha$ may not be guaranteed.**
>
> (R1) The updated Tab.3 demonstrate that finding a good enough $\alpha$ takes small cost. Most of a good $\alpha$ require only one run. We aim to find a better $\alpha$ instead of the most optimal as it outperforms the original generation. Please refer to A1 to Reviewer sVfQ (part 2/4) for a detail answer.
>
> >**(W2) The paper does not make a comparison with other modulation methods. For example, the following works also propose feature modulation methods, although the focus is towards high-resolution images.**
>
> (R2) We have added comparisons on T2I-CompBench against two benchmarks, FreeU and Attend-and-Excite. All settings follow the official default code. We report the results in Sec.4.4 ("Evaluation on T2I-CompBench") and Tab.4 as shown below. Concretely, we report BLIP-VQA correctness scores for attribute binding (color, shape, texture), UniDet-based detection scores for object presence and spatial relations, and CLIP-based text--image similarity scores, which are further aggregated into the “Numeracy”, “3D”, “Non-spatial”, and “Spatial” composite metrics shown in Tab.4.
>
> | Method            | Backbone | Color         | Shape         | Texture       | Numeracy       | 3D            | Non-spat.      | Spat.         |
> |-------------------|----------|---------------|---------------|---------------|----------------|---------------|----------------|---------------|
> | Baseline          | SD2.1    | 0.40±0.35     | 0.39±0.30     | 0.47±0.32     | 0.34±0.23      | 0.23±0.16     | 0.30±0.03      | 0.05±0.21     |
> |                   | SDXL     | 0.60±0.36     | 0.48±0.31     | 0.54±0.37     | 0.49±0.24      | 0.37±0.18     | 0.31±0.03      | 0.23±0.40     |
> | FreeU             | SD2.1    | 0.58±0.14     | 0.57±0.14     | 0.55±0.15     | 0.46±0.27      | 0.34±0.17     | 0.31±0.03      | 0.19±0.37     |
> |                   | SDXL     | 0.57±0.15     | 0.52±0.15     | 0.51±0.14     | 0.44±0.25      | 0.31±0.03     | 0.31±0.03      | 0.22±0.39     |
> | Attend-and-Excite | SD2.1    | 0.49±0.15     | 0.48±0.15     | 0.47±0.15     | 0.45±0.25      | 0.33±0.18     | 0.31±0.03      | 0.18±0.36     |
> | FM                | SD2.1    | 0.58±0.33     | 0.58±0.30     | 0.58±0.32     | 0.49±0.23      | 0.37±0.17     | 0.31±0.03      | 0.21±0.30     |
> |                   | SDXL     | **0.73±0.30** | **0.62±0.30** | **0.68±0.34** | **0.63±0.25**  | **0.48±0.20** | **0.32±0.03**  | **0.31±0.44** |
>
> The references provided by the Reviewer are similar to the baseline methods we used in Sec 4.4. We have also shown that FM works across resolutions in Appendix G. Please refer to A2 for a detail answer.
>
> >**(W3) Section 2, related work, does not explain how it is related to the proposed work or how it differs from the proposed method.**
>
> (R3) Section 2 first reviews representative text-to-image diffusion backbones, including Stable Diffusion variants and DiT-based models, to situate our setting and explain why we evaluate FM on SD2.1, SDXL, and SD3.0. It then groups control and training-free enhancement methods that also modify or guide diffusion features, and positions FM as a simple training-free feature modulation that directly acts on internal channels with a single scalar $\alpha$ for a broader modulation purpose compared to the exsiting SOTA methods (e.g., ControlNet, X-Adapter, FreeU, RB-Modulation, Make-it-Count, Attend-and-Excite...).
>
> >**(Minor) Format and grammar problems**
>
> (R4) Thank you for your careful review. We improve our paper by the following:
> - We have correctly place all table captions above the Tables.
> - Thank you & We have corrected this mistake.
> - Fig.1 is now correctly cited in line 84.
> - Thank you & We have updated this in line 499 (revised version).
> - We update the subsection name of Sec 4.1 into "Experimental Setup"

---

> ### Author Response · Authors · 2025-12-02
> **Response to Reviewer usGn (Part 2/2)**
>
> >**(Q1) What is the resolution of the images in the experiments?**
>
> (A1) All experiments use the default resolution for each model: SD2.1 at $512{\times}512$, and SDXL and SD3.0 at $1024{\times}1024$ (summarized in the Appendix L "Experimental Settings", new text in blue).
>
> >**(Q2) What is the performance of FM if the resolution varies?**
>
> (A2) FM works across resolutions. We have added multi-resolution experimental results for SDXL in the Appendix G "Experiments on Different Resolutions" (new text in blue), with representative examples shown in Fig.G12 and G13. Optimal results may require retuning parameters such as the FM channel ratio $\gamma$.
>
> >**(Q3) What is the performance comparison with other modulation methods in terms of the criteria used in the paper?**
>
> (A3) We added visual comparisons using same prompt from Fig.7 and Fig.8 with two baseline methods, FreeU and Attend-and-Excite in Appendix J and Fig.J17. We also added a comprehensive comparison on T2I-CompBench in Sec.4.4 ("Evaluation on T2I-CompBench", new text in blue) and Tab.4 as shown below:
>
> | Method            | Backbone | Color         | Shape         | Texture       | Numeracy       | 3D            | Non-spat.      | Spat.         |
> |-------------------|----------|---------------|---------------|---------------|----------------|---------------|----------------|---------------|
> | Baseline          | SD2.1    | 0.40±0.35     | 0.39±0.30     | 0.47±0.32     | 0.34±0.23      | 0.23±0.16     | 0.30±0.03      | 0.05±0.21     |
> |                   | SDXL     | 0.60±0.36     | 0.48±0.31     | 0.54±0.37     | 0.49±0.24      | 0.37±0.18     | 0.31±0.03      | 0.23±0.40     |
> | FreeU             | SD2.1    | 0.58±0.14     | 0.57±0.14     | 0.55±0.15     | 0.46±0.27      | 0.34±0.17     | 0.31±0.03      | 0.19±0.37     |
> |                   | SDXL     | 0.57±0.15     | 0.52±0.15     | 0.51±0.14     | 0.44±0.25      | 0.31±0.03     | 0.31±0.03      | 0.22±0.39     |
> | Attend-and-Excite | SD2.1    | 0.49±0.15     | 0.48±0.15     | 0.47±0.15     | 0.45±0.25      | 0.33±0.18     | 0.31±0.03      | 0.18±0.36     |
> | FM                | SD2.1    | 0.58±0.33     | 0.58±0.30     | 0.58±0.32     | 0.49±0.23      | 0.37±0.17     | 0.31±0.03      | 0.21±0.30     |
> |                   | SDXL     | **0.73±0.30** | **0.62±0.30** | **0.68±0.34** | **0.63±0.25**  | **0.48±0.20** | **0.32±0.03**  | **0.31±0.44** |
>
> Concretely, we report BLIP-VQA correctness scores for attribute binding (color, shape, texture), UniDet-based detection scores for object presence and spatial relations, and CLIP-based text--image similarity scores, which are further aggregated into the “Numeracy”, “3D”, “Non-spatial”, and “Spatial” composite metrics shown in Tab.4 and Tab.5. FM achieves state-of-the-art performance.

---

### Official Review · Reviewer_D6bC · 2025-10-30

**Soundness:** 3
**Presentation:** 3
**Contribution:** 2
**Rating:** 6
**Confidence:** 5

**Summary:**

The paper proposes **Feature Modulation** (FM), a training-free and architecture-agnostic approach for improving the qualitative fidelity and text–image alignment in text-to-image diffusion models. FM introduces a lightweight, deterministic transformation of selected feature maps during inference, without modifying model parameters. Qualitative evaluations demonstrate consistent improvements in object integrity, completeness, and numerical accuracy compared to the baseline models.

**Strengths:**

1. **Architecture-agnostic and training-free:**
   The proposed approach is architecture-agnostic, training-free, and easy to implement.
   It can, in principle, be integrated into any text-to-image diffusion model that provides access to intermediate feature maps during inference (e.g., in U-Net- or Transformer-based frameworks).

2. **Clarity and reproducibility:**
   The method is clearly described in Section 3 and can be easily understood and implemented based on the provided explanation.

3. **Comprehensive experimental coverage:**
   The experiments evaluate three Stable Diffusion models (SD 2.1, SDXL, SD 3.0), spanning two architectural families (U-Net and DiT), and three prompt datasets (DrawBench, COCO, and a small custom set). Results are primarily qualitative, based on an LLM-based evaluation, showing improvement rates between 57 % and 73 % with a very low degradation rate (< 2 %). The LLM evaluation was validated beforehand with approximately 88 % agreement with human judgments. Prompt dependency and the influence of the modulation parameter $\alpha$ were systematically analyzed in an ablation study.

**Weaknesses:**

1. **LLM-based evaluation:**
   The evaluation relies exclusively on qualitative, GPT-4o-based judgments without quantitative metrics in the main paper.
   CLIP score results provided in the appendix show small but consistent gains (at fixed $\alpha$), indicating that parameter tuning may not be strictly necessary. However, these results lack scale information and statistical significance testing, and should therefore be interpreted with caution.

2. **Hidden computational cost:**
   The best qualitative results require a grid search over the modulation parameter $\alpha$, which weakens the claim of achieving improvements “without incurring any additional computational cost.”

3. **Lack of comparison with related training-free methods:**
   No comparisons are provided to other training-free enhancement approaches (e.g., FreeU). It also remains unclear whether FM can be combined with these techniques or if they interfere with each other.

4. **Lack of theoretical analysis:**
   The paper provides no mathematical analysis of how the quadratic modulation function affects the denoising field. The motivation remains purely heuristic.

5. **Limited reproducibility:**
   Since evaluation depends on GPT-4o, the reported results are only partially reproducible.

**Questions:**

See weaknesses.

---

> ### Author Response · Authors · 2025-11-26
> **Response to Reviewer D6bC (Part 1/2)**
>
> Dear Reviewer D6bC,
>
> Thank you for your comments. We would like to answer your questions as below:
>
> >**(Q1) LLM-based evaluation: The evaluation relies exclusively on qualitative, GPT-4o-based judgments without quantitative metrics in the main paper. CLIP score results provided in the appendix show small but consistent gains (at fixed), indicating that parameter tuning may not be strictly necessary. However, these results lack scale information and statistical significance testing, and should therefore be interpreted with caution.**
>
> (A1) Besides the GPT-4o-based evaluation, we now add a conventional comparison on T2I-CompBench benchmark. Following the official code and default settings of T2I-Compbench, we evaluate the baselines, FreeU and Attend-and-Excite on SD2.1 and SDXL in Sec.4.4 ("Evaluation on T2I-CompBench"; new text in blue) and Tab.4. Concretely, we report BLIP-VQA correctness scores for attribute binding (color, shape, texture), UniDet-based detection scores for object presence and spatial relations, and CLIP-based text--image similarity scores, which are further aggregated into the “Numeracy”, “3D”, “Non-spatial”, and “Spatial” composite metrics shown in Tab.4 and Tab.5.
>
> >**(Q2) Hidden computational cost: The best qualitative results require a grid search over the modulation parameter $\alpha$, which weakens the claim of achieving improvements “without incurring any additional computational cost.”**
>
> (A2) We have clarified in Sec.5 that FM requires no extra "training" cost. Our claim of “no extra cost” is intended in the training sense: FM is a training-free method that does not introduce any additional parameters or training stages. We admit that requiring some additional sampling cost is one of the limitations. However, this cost is small and controlled. As detailed in Sec.4, we restrict $\alpha$ to a very small grid $\{-0.2,-0.1,0.1,0.2\}$ and use early stopping, so each prompt requires at most four FM samples beyond the baseline, and in practice most improved prompts are achieved after one or two trials. Tab.3 summarizes, for each dataset and search setting, the frequency of each $\alpha$ is the first value that yields an improvement under LLM evaluation, quantitatively showing that $\alpha=-0.2$ or $\alpha=-0.1$ already works for the majority of cases.
>
> >**(Q3) Lack of comparison with related training-free methods: No comparisons are provided to other training-free enhancement approaches (e.g., FreeU). It also remains unclear whether FM can be combined with these techniques or if they interfere with each other.**
>
> (A3) We have added comparisons on T2I-CompBench against two benchmarks, FreeU and Attend-and-Excite. All settings follow the official default code. We report the results in Sec.4.4 ("Evaluation on T2I-CompBench") and Tab.4 as shown below. Conceptually, FM operates as a plug-and-play modulation on intermediate feature maps and does not conflict in principle with methods such as FreeU or Attend-and-Excite. In practice, jointly tuning multiple training-free modules would introduce many additional hyperparameters and make the combination strategy more complex. We leave systematic combinations as future work.
>
> | Method            | Backbone | Color         | Shape         | Texture       | Numeracy       | 3D            | Non-spat.      | Spat.         |
> |-------------------|----------|---------------|---------------|---------------|----------------|---------------|----------------|---------------|
> | Baseline          | SD2.1    | 0.40±0.35     | 0.39±0.30     | 0.47±0.32     | 0.34±0.23      | 0.23±0.16     | 0.30±0.03      | 0.05±0.21     |
> |                   | SDXL     | 0.60±0.36     | 0.48±0.31     | 0.54±0.37     | 0.49±0.24      | 0.37±0.18     | 0.31±0.03      | 0.23±0.40     |
> | FreeU             | SD2.1    | 0.58±0.14     | 0.57±0.14     | 0.55±0.15     | 0.46±0.27      | 0.34±0.17     | 0.31±0.03      | 0.19±0.37     |
> |                   | SDXL     | 0.57±0.15     | 0.52±0.15     | 0.51±0.14     | 0.44±0.25      | 0.31±0.03     | 0.31±0.03      | 0.22±0.39     |
> | Attend-and-Excite | SD2.1    | 0.49±0.15     | 0.48±0.15     | 0.47±0.15     | 0.45±0.25      | 0.33±0.18     | 0.31±0.03      | 0.18±0.36     |
> | FM                | SD2.1    | 0.58±0.33     | 0.58±0.30     | 0.58±0.32     | 0.49±0.23      | 0.37±0.17     | 0.31±0.03      | 0.21±0.30     |
> |                   | SDXL     | **0.73±0.30** | **0.62±0.30** | **0.68±0.34** | **0.63±0.25**  | **0.48±0.20** | **0.32±0.03**  | **0.31±0.44** |

---

> ### Author Response · Authors · 2025-11-26
> **Response to Reviewer D6bC (Part 2/2)**
>
> >**(Q4) Lack of theoretical analysis: The paper provides no mathematical analysis of how the quadratic modulation function affects the denoising field. The motivation remains purely heuristic.**
>
> (A4) We have added the Appendix C for explaining the Effect on the Denoising Field. There we view the denoising network at each timestep $t$ as defining a time-dependent vector field $F(x_t,t)$ that maps the noisy latent $x_t$ to its denoised update. In our encoder–decoder decomposition, the original (unmodulated) field is written as
> $F_0(x_t,t) = f_{\mathrm{dec}}(f_{\mathrm{enc}}(x_t,t),t)$,
> and applying FM to the encoder features gives
> $F_\alpha(x_t,t) = f_{\mathrm{dec}}(f_M(f_{\mathrm{enc}}(x_t,t);\alpha),t)$,
> so that the change in the field is
> $\Delta F(x_t,t;\alpha) = F_\alpha(x_t,t) - F_0(x_t,t)$.
> For moderate $\alpha$, we then perform a first-order Taylor expansion around $\alpha=0$ and obtain
> $F_\alpha(x_t,t) \approx F_0(x_t,t) + \mathcal{A}(x_t,t)\,\alpha$,
> where $\mathcal{A}(x_t,t)$ collects the decoder sensitivity to the selected top-$\gamma$ channels together with the derivative of $f_M$ with respect to $\alpha$. In other words, FM acts as a small, approximately linear perturbation of the reverse-time denoising field along dominant semantic feature directions. In Appendix E, visualizations of the FM effect server as an additional analysis on how the FM affect the feature map.
>
> >**(Q5) Limited reproducibility: Since evaluation depends on GPT-4o, the reported results are only partially reproducible.**
>
> (A5) Beyond the current GPT-4o evaluation, we add the T2I-CompBench evaluation using only the official benchmark code and default settings for SD2.1 and SDXL. Our generated images and results on all experiments are fully reproducible with the seeds reported in our paper. In addition, after the review period we will release our entire code (including the code already submitted and the new T2I-CompBench runs) to ensure reproducibility.

---

### Official Review · Reviewer_M9ec · 2025-10-31

**Soundness:** 2
**Presentation:** 2
**Contribution:** 3
**Rating:** 6
**Confidence:** 3

**Summary:**

This paper introduces Feature Modulating (FM), a training-free method designed to enhance image quality and text–image alignment in text-to-image diffusion models. Instead of using extra inputs or retraining, FM directly modulates latent features during the denoising process. The authors design a channel-wise monotonic modulation function controlled by a single parameter to adjust feature values efficiently. This modulation alters the internal feature distribution, leading to improved generation quality. The proposed FM module is architecture-agnostic and can be easily integrated into existing diffusion frameworks. Experiments on multiple benchmarks show consistent improvements in image quality, semantic fidelity, and realism.

**Strengths:**

1. This paper is well-structured and easy to follow.

2. The proposed feature modulating approach appears novel.

**Weaknesses:**

1. The authors propose a quadratic modulation strategy that reshapes the feature distribution while preserving the monotonic ordering of feature values. The modulation effect is formalized in the lemma. Could the authors provide a simulation experiment or distribution example to intuitively illustrate this modulation effect?

2. In Lines 212–213, it is stated that “negative $\alpha$ values can potentially help pull back out-of-distribution features.” What exactly are out-of-distribution features in this context? Please describe their impact and underlying mechanism in detail.

3. For the experiments on SD2.1 and SDXL, which layer is selected to apply the FM module?

4. For the experiments on SD2.1 and SDXL, $\gamma$ is set to 0.1, and for SD3.0, $\gamma$ is set to 0.5. How was the optimal $\gamma$ determined? Was it chosen based on quantitative experimental observation? Please include ablation studies on $\gamma$ to support this choice.

5. As mentioned, “the optimal setting of $\alpha$ is highly image-dependent and varies significantly across different cases.” On average, how many search steps are required for each image generation? Please provide ablation studies on $\alpha$.

6. Could the authors provide practical guidance or a recommended range of $\alpha$ values for different generation tasks to facilitate reproducibility and application?

**Questions:**

See Weaknesses.

---

> ### Author Response · Authors · 2025-11-26
> **Response to Reviewer M9ec**
>
> Dear Reviewer M9ec,
>
> Thank you very much for your insightful comments. We would like to answer your questions by following:
>
> >**(Q1) The authors propose a quadratic modulation strategy that reshapes the feature distribution while preserving the monotonic ordering of feature values. The modulation effect is formalized in the lemma. Could the authors provide a simulation experiment or distribution example to intuitively illustrate this modulation effect?**
>
> (A1) How our FM function preserves the monotonic ordering is breifly presented in Fig.3 (c). We have further added the simulation experiment on how the positive and negative alpha affect the feature maps along the channel demension in the new Appendix E in Fig.E9 and Fig.E10, together with the visualization of the distribution change.
>
> >**(Q2) In Lines 212–213, it is stated that “negative values can potentially help pull back out-of-distribution features.” What exactly are out-of-distribution features in this context? Please describe their impact and underlying mechanism in detail.**
>
> (A2) Thank you very much for bringing this question. We agree that our original wording was imprecise. What we mean "out-of-distribution" are more like *outlier* channels with potentially unusually large normalized activations within the selected top-$\gamma$ set (the cyan bins in Fig.5), rather than features that are out-of-distribution with respect to the training data. FM with negative $\alpha$ reduces the means of these channels, pulling such outliers back toward the bulk of the distribution and avoiding over-emphasis of spurious responses. We now consistently use "potential outlier features" in Sec.3 and the Sec.4.5.
>
> >**(Q3) For the experiments on SD2.1 and SDXL, which layer is selected to apply the FM module?**
>
> (A3) We provide full explaination in Appendix P on applying FM on different layers. For SD2.1 and SDXL we apply FM to intermediate decoder blocks and deep skip-connection features where high-level semantics dominate. The exact positions and an ablation over these choices are detailed in Appendix P (Fig.P19 and Figs.P20--P23).
>
> >**(Q4) For the experiments on SD2.1 and SDXL, $\gamma$ is set to 0.1, and for SD3.0, $\gamma$ is set to 0.5. How was the optimal determined? Was it chosen based on quantitative experimental observation? Please include ablation studies on to support this choice.**
>
> (A4) The channel-selection ratio $\gamma$ is chosen from quantitative ablations reported in Appendix O (Fig.O18). For SD2.1 and SDXL, $\gamma=0.1$ provides a good balance between effectiveness and stability, while for SD3.0, where FM is applied closer to the output of a DiT backbone, a larger $\gamma=0.5$ is more effective.
>
> >**(Q5) As mentioned, “the optimal setting of $\alpha$ is highly image-dependent and varies significantly across different cases.” On average, how many search steps are required for each image generation? Please provide ablation studies on $\alpha$.**
>
> (A5) We have update the original words in Sec 4.3 for a better description on the $\alpha$ by "Since the optimal setting of $\alpha$ may vary from image to image, even though multiple $\alpha$ values can improve a given example, conducting a strictly fair statistical evaluation (such as CLIP score, please see the Appendix) remains non-trivial." In practice, we treat $\alpha$ as a scalar control knob and search over a very small grid, typically $\alpha\in\{-0.2,-0.1,0.1,0.2\}$ in a fixed order from $-0.2$ toward $0.2$; in most cases one or two non-zero values already improve the image, so only a few extra FM samples per prompt are needed. The newly addded Tab 3 and Tab.5 provide the selection on $\alpha$,
>
> Sec.4.4 and Fig.8 provide an ablation on $\alpha$, showing that values in the range $[-0.2,0.2]$ offer a good trade-off between correcting failures (e.g., object integrity and counting) and preserving the original composition, while larger $|\alpha|$ tends to over-modify the image. In addition, the T2I-CompBench analysis in Sec.4 reports the frequency of each candidate $\alpha$ is selected as the per-image best value for SD2.1 and SDXL, quantitatively confirming that effective settings are concentrated in this small grid rather than requiring a wide or fine-grained search.
>
> >**(Q6) Could the authors provide practical guidance or a recommended range of $\alpha$ values for different generation tasks to facilitate reproducibility and application?**
>
> (A6) Consistent with the observations reported in the above answer, our experiments suggest that a small range around zero already works well in practice. For most generation tasks we recommend restricting $\alpha$ to $[-0.2,0.2]$ of all testing baseline models: small positive values (e.g., $\alpha=0.1$ or $0.2$) are effective for improving object integrity and counting, while small negative values (e.g., $\alpha=-0.1$ or $-0.2$) are better for sharpening local textures or revealing suppressed fine details. We summarize this guidance in Sec.4.5 last paragraph.

---

### Official Review · Reviewer_hMb4 · 2025-11-01

**Soundness:** 2
**Presentation:** 3
**Contribution:** 2
**Rating:** 4
**Confidence:** 3

**Summary:**

This paper introduces a training-free Feature Modulation (FM) method designed to enhance the performance of diffusion models. The proposed method first selects a fraction $\gamma$ of feature channels exhibiting the highest mean activations. These selected channels are then modulated by a quadratic function, which is controlled by a parameter $\alpha$. Experiments on both UNet and DiT-based diffusion models demonstrate that the proposed method improves integrity, numerical accuracy, completeness, and visual refinement.

**Strengths:**

1. The proposed method is training-free, requiring no extra training or fine-tuning, and is shown to be effective across both UNet and DiT architectures.
2. The writing is clear and easy to follow.

**Weaknesses:**

1. As mentioned in Line 384, the modulation parameter $\alpha$ appears to require careful tuning for each specific case, which may introduce additional computational overhead and inconvenience in practice.
2. Most of the ablation studies (including Section 4.4, Appendix I and J) relies on empirical visualizations, which may only reflect a limited set of cases. It would be better to include statistical or quantitative analyses to support broader, more generalizable conclusions.

**Questions:**

1. Are the modulation be applied to all timesteps throughout the sampling process? Is it possible to use different $\alpha$ for different timesteps?
2. In the first row of Figure 8, both increasing and decreasing $\alpha$ increase the number of generated dogs. Could the authors clarify why both positive and negative $\alpha$ values lead to similar effects in this example?

---

> ### Author Response · Authors · 2025-11-26
> **Response to Reviewer hMb4 (Part 1/2)**
>
> Dear Reviewer hMb4,
>
> Thank you for recognizing the clarity of our paper. We answer your questions one by one:
>
> > **(W1) As mentioned in Line 384, the modulation parameter $\alpha$ appears to require careful tuning for each specific case, which may introduce additional computational overhead and inconvenience in practice.**
>
> (R1) We add Tab.3 in Sec 4.3 to demonstrate that finding a good $\alpha$ cost little. The table below summarizes, for each dataset in Tab.2 and for both search settings (two $\alpha$ values vs. four), how often each $\alpha$ is the first value that yields an improvement under GPT‑4o:
>
>
>   | Dataset   | $\alpha$ set                  |   -0.2           |   -0.1           |    0.1           |    0.2           |
>   |-----------|------------------------------|------------------|------------------|------------------|------------------|
>   | Custom    | $\{-0.2,\,0.2\}$             | 17 (85.00\%)     |   –              |   –              |  3 (15.00\%)     |
>   | Custom    | $\{-0.2,-0.1,0.1,0.2\}$      | 15 (62.50\%)     |  6 (25.00\%)     |  2 ( 8.33\%)     |  1 ( 4.17\%)     |
>   | DrawBench | $\{-0.2,\,0.2\}$             | 130 (84.97\%)    |   –              |   –              | 23 (15.03\%)     |
>   | DrawBench | $\{-0.2,-0.1,0.1,0.2\}$      | 116 (69.46\%)    | 26 (15.57\%)     | 15 ( 8.98\%)     | 10 ( 5.99\%)     |
>   | COCO      | $\{-0.2,\,0.2\}$             | 355 (66.85\%)    |   –              |   –              | 176 (33.15\%)    |
>   | COCO      | $\{-0.2,-0.1,0.1,0.2\}$      | 367 (52.28\%)    | 156 (22.22\%)    | 128 (18.23\%)    | 51 ( 7.26\%)     |
>
> Most of the runs achieve a better result in the first attempt on searching $\alpha$. We also report the computational overhead and and win‑rate vs. cost trade‑off in A1 to Reviewer sVfQ (Part 2/4).
>
>
> > **(W2) Most of the ablation studies (including Section 4.4, Appendix I and J) relies on empirical visualizations, which may only reflect a limited set of cases. It would be better to include statistical or quantitative analyses to support broader, more generalizable conclusions.**
>
> (R2) We now provide several complementary quantitative analyses. This includes the new Sec 4.4, Tab.3, Tab.4 Tab.5 and Appendix E. We add a T2I-CompBench evaluation in Sec 4.4 and Tab 4 where FM is compared side-by-side with the baselines. In Tab.3 and Tab.5 we provide the seaching cost of $\alpha\in\{-0.2,-0.1,0.1,0.2\}$. The appendix E shows more analytics of FM’s channel-wise impact on feature maps. We present the Tab 4 with the main results of T2I-Compbench Evaluation here:
>
> | Method            | Backbone | Color         | Shape         | Texture       | Numeracy       | 3D            | Non-spat.      | Spat.         |
> |-------------------|----------|---------------|---------------|---------------|----------------|---------------|----------------|---------------|
> | Baseline          | SD2.1    | 0.40±0.35     | 0.39±0.30     | 0.47±0.32     | 0.34±0.23      | 0.23±0.16     | 0.30±0.03      | 0.05±0.21     |
> |                   | SDXL     | 0.60±0.36     | 0.48±0.31     | 0.54±0.37     | 0.49±0.24      | 0.37±0.18     | 0.31±0.03      | 0.23±0.40     |
> | FreeU             | SD2.1    | 0.58±0.14     | 0.57±0.14     | 0.55±0.15     | 0.46±0.27      | 0.34±0.17     | 0.31±0.03      | 0.19±0.37     |
> |                   | SDXL     | 0.57±0.15     | 0.52±0.15     | 0.51±0.14     | 0.44±0.25      | 0.31±0.03     | 0.31±0.03      | 0.22±0.39     |
> | Attend-and-Excite | SD2.1    | 0.49±0.15     | 0.48±0.15     | 0.47±0.15     | 0.45±0.25      | 0.33±0.18     | 0.31±0.03      | 0.18±0.36     |
> | FM                | SD2.1    | 0.58±0.33     | 0.58±0.30     | 0.58±0.32     | 0.49±0.23      | 0.37±0.17     | 0.31±0.03      | 0.21±0.30     |
> |                   | SDXL     | **0.73±0.30** | **0.62±0.30** | **0.68±0.34** | **0.63±0.25**  | **0.48±0.20** | **0.32±0.03**  | **0.31±0.44** |
>
> Concretely, we report BLIP-VQA correctness scores for attribute binding (color, shape, texture), UniDet-based detection scores for object presence and spatial relations, and CLIP-based text--image similarity scores, which are further aggregated into the “Numeracy”, “3D”, “Non-spatial”, and “Spatial” composite metrics shown in Tab.4

---

> ### Author Response · Authors · 2025-12-02
> **Response to Reviewer hMb4 (Part 2/2)**
>
> >**(Q1) Are the modulation be applied to all timesteps throughout the sampling process? Is it possible to use different for different timesteps?**
>
> (A1) Yes. In our default implementation, FM is applied to all denoising timesteps with a single global modulation strength $\alpha$, which already gives strong improvements and keeps the method simple and consistent to use. For this revision, we additionally
> - add an explicit sentence at the end of Sec.3 (Methodology, Feature Modulation) stating that we use a single global $\alpha$ across all denoising steps.
> - add an appendix F "Different FM for Different Steps" where we visualize variants with different $\alpha$ across timesteps.
>
> >**(Q2) In the first row of Figure 8, both increasing and decreasing $\alpha$ increase the number of generated dogs. Could the authors clarify why both positive and negative values lead to similar effects in this example?**
>
> (A2) We would like to express our sincere thanks and apologies for this careful observation. After carefully reviewing our code and results, we found that the first row of Fig.8 (the "five dogs" example) was assembled incorrectly: the results are mixed with an earlier version of the code. We have corrected Fig.8 in the main paper and, for full transparency, added a dedicated appendix I with the complete grids for this prompt , showing all $\alpha$ values and layers for SDXL. In the final camera-ready version, this clarifying appendix section will be removed, leaving only the corrected main-figure row. The entire code will be released after double-blind review period to make sure full reproducibility. Thank you again for your careful and thorough review.

---

### Author Response · Authors · 2025-11-26
**General Response to All Reviewers**

Dear AC and all reviewers,

We thank all reviewers for their insightful and constructive comments. Several concerns recur across the reviews, especially regarding the need for more statistical evaluation and clarification of our experimental design. We briefly summarize our main revisions here before addressing each reviewer individually. Because most added experiments are presented as figures, please see the revised PDF: all newly added content is highlighted in blue and deleted content in red. We respond to two frequently raised points below:

1. **New Sec.4.4 for Additional benchmark: T2I-CompBench.** We added an evaluation on T2I-CompBench [1] with the latest T2I-CompBench++ version [2] using the official benchmark code and default settings. We compare FM with FreeU [3] and Attend-and-Excite [4] on SD2.1 and SDXL. T2I-CompBench serves as a multi-dimensional quantitative evaluation combining automatic metrics (BLIP-VQA, UniDet, CLIPScore, etc.) designed to assess compositional text--image alignment (color, shape, texture, spatial relations, numeracy). Under this setup we report compositional scores for the baselines, FreeU, Attend-and-Excite, and FM in revised Sec.4.4 ("Evaluation on T2I-CompBench") and Tab 4 (blue). A further advantage is that FM applies to the DiT-based SD3 backbone, where baseline methods are not applicable. Other baselines such as RB-Modulation or Make-It-Count overlap less directly with our setting: they target narrower objectives (e.g., style transfer or pure counting) or require additional training, and therefore are not directly comparable to our training-free, diversity-controlling modulation.

2. **New Tab.3 and Tab.5 for the “optimal” $\alpha$ selection.**
We added Tab.3 to show that sweeping a small fixed grid $\{-0.2,0.2\}$ or $\{-0.2,-0.1,0.1,0.2\}$ in order from $-0.2$ to $0.2$ finds a better $\alpha$ than the baseline image with low cost. Additional details on searching for the best $\alpha$ on T2I-CompBench are in Sec.4.4 and Tab.5. We acknowledge the extra sampling overhead as a limitation of FM. However, our focus is that a very simple, training-free, architecture-agnostic feature modulation with a single scalar $\alpha$ already yields strong gains in text--image alignment and visual quality.

3. **Appendix changes:**
- Updated Appendix C (Effect on the Denoising Field): theoretical analysis of FM’s impact on the denoising process.
- Updated Appendix E (Visualizations of the FM Effect): more examples of FM’s channel-wise impact on feature maps.
- Updated Appendix F (Different FM for Different Steps): using different $\alpha$ across timesteps.
- Updated Appendix G (Experiments on Different Resolutions): results under varying resolutions.
- Updated Appendix H (FM on More Model Architectures): examples on Flux, SANA, and Qwen.
- Updated Appendix I (Clarification of the “five dogs” example in Fig. 8): detailed correction of the original Fig. 8 error.
- Updated Appendix J (Visualization Compariason with Baseline Methods): Visualization compariason with the same prompts used in Fig.7 and Fig.8.
- Updated Appendix L (Experimental Settings): detailed settings to ensure full reproducibility.
- Updated Appendix M (Results on Different Seeds and Samplers): Results with different seeds and samplers.

The remaining concerns raised by each reviewer are addressed point-by-point in the following sections.

[1] Huang, K., Sun, K., Xie, E., Li, Z., & Liu, X. (2023). T2I-CompBench: A comprehensive benchmark for open-world compositional text-to-image generation. *NeurIPS* 36, 78723–78747.
[2] Huang, K., Duan, C., Sun, K., Xie, E., Li, Z., & Liu, X. (2025). T2I-CompBench++: An enhanced and comprehensive benchmark for compositional text-to-image generation. *IEEE TPAMI*, 1–17.
[3] Si, C., Huang, Z., Jiang, Y., & Liu, Z. (2024). FreeU: Free lunch in diffusion U-Net. In *CVPR*.
[4] Chefer, H., Alaluf, Y., Vinker, Y., Wolf, L., & Cohen-Or, D. (2023). Attend-and-Excite: Attention-based semantic guidance for text-to-image diffusion models. *ACM Transactions on Graphics*, 42(4), 148:1–148:10.

---

### Author Response · Authors · 2025-12-03
**Review and Reviewer-Author Discussion Summary (1/3)**

Dear PCs, SACs, ACs, and Reviewers,

Thank you very much for your valuable contributions to our work. To assist the newly assigned AC and help reduce their workload, we provide below a summary of the key positive points from the reviews.

### Strengths

Overall, we are grateful that the reviewers identified several aspects of our work:

1. **Our FM module provides a new, simple, training-free, architecture-agnostic channel-wise mechanism that can be easily plugged into both U‑Net‑ and DiT‑based diffusion models.**
All six reviewers recognized this point (hMb4: Strength 1, M9ec: Strength 2, D6bC: Strength 1, usGn: Strength 1, sVfQ: Strength 1, rTb9: Strength 1).

2. **The paper is clearly written, well-structured, and the FM module is easy to understand and reimplement.**
Three reviewers recognized this point (hMb4: Strength 2, M9ec: Strength 1, D6bC: Strength 2).

3. **FM consistently improves image quality, object integrity and completeness, numerical accuracy, and text–image alignment across multiple models and prompt sets.**
Three reviewers recognized this point (D6bC: Strength 3, usGn: Strength 2, sVfQ: Strength 2).

4. **The experiments provide comprehensive coverage of different backbones and datasets, together with systematic analyses such as prompt dependency and the effect of the modulation parameter.**
Two reviewers recognized this point (D6bC: Strength 3, sVfQ: Strength 2).

5. **The paper adopts a practical LLM-based evaluation protocol validated against human judgments, with consistent improvement rates and low degradation.**
Two reviewers recognized this point (D6bC: Strength 3, usGn: Strength 3).

---

> ### Author Response · Authors · 2025-12-03
> **Review and Reviewer-Author Discussion Summary (2/3)**
>
> ### Concerns and Our Addressing
>
> During the discussion and revision period, reviewers raised several recurring concerns, mainly about evaluation rigor, hyperparameter tuning, comparisons to related methods, theoretical intuition, generality, and reproducibility. We briefly summarize how the current version addresses these points:
>
> 1. **Evaluation rigor and reliance on GPT‑4o.** Reviewers asked for stronger quantitative evidence beyond our GPT‑4o–based protocol. In response, we added a new Sec. 4.4 and Tabs. 4–5 with T2I‑CompBench/T2I‑CompBench++ results on SD2.1 and SDXL, reporting BLIP‑VQA, UniDet, and CLIP-based compositional scores and comparing FM with FreeU and Attend-and-Excite under the official evaluation pipeline. FM consistently improves over the baselines and is competitive with or better than these training‑free methods, and Appendix M further shows that the gains are robust across seeds and samplers. This directly addresses the evaluation-related concerns of reviewers **hMb4** (Weakness 2), **D6bC** (Weaknesses 1 and 5), **sVfQ** (Weakness 1), **rTb9** (Weakness 3 and Question 3), and **usGn** (Question 3).
> 2. **Hyperparameter $\alpha$, computational overhead, and practical usability.** Reviewers were concerned that per‑prompt tuning of $\alpha$ could be expensive and make FM hard to use in practice. We added Tabs. 3 and 5 and new text in Sec. 4.3–4.5 to show that a tiny discrete grid (e.g., $\{-0.2,-0.1,0.1,0.2\}$ with early stopping) suffices: most prompts are improved by the first one or two values, and at most four extra FM samples per prompt are needed. We also report FLOPs and wall‑clock overhead and summarize clear small searching grid for choosing $\alpha$, so that it functions as a simple user‑controlled knob rather than a fragile hyperparameter. This responds to concerns from **hMb4** (Weakness 1), **M9ec** (Weaknesses 4–6), **D6bC** (Weakness 2), **usGn** (Weakness 1), **sVfQ** (Weakness 2 and Question 1), and **rTb9** (Weakness 2 and Question 2).
> 3. **Comparison with related training‑free methods.** Reviewers requested more direct comparisons to existing training‑free modulation approaches such as FreeU, Attend-and-Excite, RB‑Modulation, and Make‑It‑Count. The revised Sec. 4.4 now provides a systematic comparison with FreeU and Attend-and-Excite on T2I‑CompBench using identical prompts and metrics. Appendix J supplements this with side‑by‑side visualizations on the same prompts as Fig. 7–8. These changes directly target the comparison-related concerns of **D6bC** (Weakness 3), **usGn** (Weakness 2 and Question 3), **sVfQ** (Weakness 3 and Question 2), and **rTb9** (Weakness 3 and Question 3).

---

> ### Author Response · Authors · 2025-12-03
> **Review and Reviewer-Author Discussion Summary (3/3)**
>
> 4. **Theoretical understanding and intuition for FM.** To address requests for a deeper explanation of why the quadratic modulation works and how it interacts with the denoising process, we substantially expanded Appendix C (“Effect on the Denoising Field”) with a mathematical analysis showing how FM reshapes the denoising vector field while preserving feature ordering. Appendix E adds simulation examples and feature‑distribution plots that visualize FM’s effect on channel activations and clarify our notion of “out‑of‑distribution” features, explaining, for example, why small negative $\alpha$ values can regularize certain activations. This addresses the theory/intuition concerns of **M9ec** (Weaknesses 1–2), **D6bC** (Weakness 4), and **hMb4** (Question 2).
> 5. **Generality across architectures, resolutions, and timesteps (including SD3.0).** Reviewers questioned whether FM is truly architecture‑agnostic and robust in different settings, and why early‑layer modulation sometimes “crashes” SD3.0. Appendix H now reports experiments on additional backbones (Flux, SANA, Qwen‑Image), where FM continues to improve image quality and compositional alignment, supporting our generality claim. Appendix G adds multi‑resolution experiments, Appendix F studies different $\alpha$ across timesteps, and we analyze SD3.0’s behavior, showing that restricting FM to the final MM‑DiT block avoids instability while still giving strong gains and discussing why simply shrinking the channel‑selection ratio $\gamma$ is insufficient. These results respond to concerns from **rTb9** (Weakness 1 and Question 1), **sVfQ** (Weakness 3 and Question 4), and **usGn** (Questions 1–2).
> 6. **Implementation details and reproducibility.** Finally, reviewers requested clearer implementation details and reassurance about reproducibility. In the main text and Appendix G, L, N and R, we specify the default resolutions, the full GPT‑4o prompt protocol, and exactly which layers and timesteps are modulated for each backbone and the recommended $\alpha$ grid. Importantly, the new T2I‑CompBench experiments rely only on public code and models, we run the baselines using all default settings from the official code. Our code is already submitted in the Supplementary Material. We will further publish our entire code publicly to ensure full reproducibility. This addresses implementation and reproducibility concerns from **hMb4** (Question 1), **M9ec** (Weaknesses 3–4 and 6), **D6bC** (Weakness 5), and **usGn** (Weakness 3 and related minor comments).
>
> We hope this concise summary of concerns and our corresponding revisions, together with the strengths listed above, helps the newly assigned AC quickly assess the current version. We are deeply grateful to the reviewers, ACs, SACs, and PCs for their careful evaluation and constructive feedback, which have substantially strengthened the paper.
>
> Sincerely,
> Authors of Submission 8897

---

### Meta-Review · Area_Chair_5VCn · 2026-01-08

**Summary:**

This paper aims to address the significant problem of improving image quality and text-image alignment in text-to-image diffusion models. Despite the importance of this task, the reviews were mixed: four reviewers gave negative scores, while only two were satisfied with the results.

**Reviewer Concerns:**

The primary concerns from reviewers involved the methodology, experimental design, and evaluation. While the authors responded to these points in the rebuttal, but it seemed not to satisfy the reviewers.

**Reviewer Scores:**

The reviewers are likely to maintain their scores.

---

### Decision · Program_Chairs · 2026-01-26

Reject